# Overparameterization from Computational Constraints

**Sanjam Garg**[*]      **Somesh Jha**[†]      **Saeed Mahloujifar**[‡]

**Mohammad Mahmoody**[§]      **Mingyuan Wang**[¶]

## Abstract

Overparameterized models with millions of parameters have been hugely successful. In this work, we ask: can the need for large models be, at least in part, due to the *computational* limitations of the learner? Additionally, we ask, is this situation exacerbated for *robust* learning? We show that this indeed could be the case. We show learning tasks for which computationally bounded learners need *significantly more* model parameters than what information-theoretic learners need. Furthermore, we show that even more model parameters could be necessary for robust learning. In particular, for computationally bounded learners, we extend the recent result of Bubeck and Sellke [NeurIPS'2021] which shows that robust models might need more parameters, to the computational regime and show that bounded learners could provably need an even larger number of parameters. Then, we address the following related question: can we hope to remedy the situation for robust computationally bounded learning by restricting *adversaries* to also be computationally bounded for sake of obtaining models with fewer parameters? Here again, we show that this could be possible. Specifically, building on the work of Garg, Jha, Mahloujifar, and Mahmoody [ALT'2020], we demonstrate a learning task that can be learned efficiently and robustly against a computationally bounded attacker, while to be robust against an information-theoretic attacker requires the learner to utilize significantly more parameters.

## 1 Introduction

In recent years, deep neural nets with millions or even billions of parameters[6] [Wortsman et al., 2022, Dai et al., 2021, Yu et al., 2022] have emerged as one of the most powerful models for very basic tasks such as image classification. A magic of DNNs is that they generalize without falling into the classical theories mentioned above, and hence they are the subject of an active line of work aiming to understand how DNNs generalize Arora et al. [2019], Allen-Zhu et al. [2019b,a], Novak et al. [2018], Neyshabur et al. [2014], Kawaguchi et al. [2017], Zhang et al. [2021], Arora et al. [2018b] and the various benefits of overparametrized regimes Xu et al. [2018], Chang et al. [2020], Arora et al. [2018a], Du et al. [2018], Lee et al. [2019]. In fact, the number of parameters in such models is so large that it is enough to memorize (and fit) to a large number of *random labels* [Zhang et al.,

---

[*]UC Berkeley and NTT Research sanjamg@berkeley.edu

[†]University of Wisconsin, Madison jha@cs.wisc.edu

[‡]Princeton University sfar@princeton.edu

[§]University of Virginia mohammad@virginia.edu

[¶]UC Berkeley mingyuan@berkeley.edu

[6]See `https://paperswithcode.com/sota/image-classification-on-imagenet` for the size of the most successful models for image classification of Imagenet.

---

36th Conference on Neural Information Processing Systems (NeurIPS 2022).

2021]. This leads us to our first main question, in which we investigate the potential cause for having large models:

> *Are there any learning tasks for which computationally bounded learners need to utilize significantly more model parameters than needed information-theoretically?*

One should be cautious in how to formulate the question above. That is because, many simple (information-theoretically learnable) tasks are believed to be computationally hard to learn (e.g., learning parity with noise [Pietrzak, 2012]). In that case, one can interpret this as saying that the efficient learner requires *infinite* number of parameters, as a way of saying that the learning is not possible at all! However, as explained above, we are interested in understanding the reason behind needing a large number of model parameters when learning *is possible*.

**Could robustness also be a cause for overparameterization?** A highly sought-after property of machine learning models is their *robustness* to the so-called adversarial examples [Goodfellow et al., 2014]. Here we would like to find a model $f$ such that $f(x) = f(x')$ holds with high probability when $x \leftarrow D$ is an honestly sampled instance and $x' \approx x$ is a *close* instance that is perhaps minimally (yet adversarially) perturbed.[7] A recent work of Bubeck and Sellke [2021] showed that having large model parameters *could* be due to the robustness of the model. Here, we are asking whether such phenomenon can have a computational variant that perhaps leads to needing *even more* parameters when the robust learner is running in polynomial time.

> *Are there any learning tasks for which computationally bounded **robust** learning comes at the cost of having even more model parameters than non-robust learning?*

In fact, it was shown by Bubeck et al. [2019] and Degwaker et al. [Degwekar et al., 2019] that computational limitations of the learner *could* be the reason behind the vulnerability of models to adversarial examples. In this work, we ask whether the phenomenon investigated by the prior works is also crucial when the model size is considered.

**Can computational intractability of adversary help?** We ask whether natural restrictions on *the adversary* can help reduce model sizes. In particular, we consider the restriction to the class of polynomially bounded adversaries. In fact, when it comes to robust learning, the computationally bounded aspect could be imposed both on the learner as well as the *adversary*.

> *Are there any learning tasks for which robust learning can be done with fewer model parameters when dealing with **polynomial-time** adversaries?*

Previously, it was shown that indeed working with *computationally bounded* adversaries *can* help achieving robust models [Mahloujifar and Mahmoody, 2019, Bubeck et al., 2019, Garg et al., 2020b]. Hence, we are asking a similar question in the context of model parameters.

## 1.1 Our results

In summary, we prove that under the computational assumption that one-way functions exist, the answer to all three of our main questions above is positive. Indeed, we show that the computational efficiency of the learner could be the cause of having overparametarized models. Furthermore, the computational efficiency of the adversary could reduce the size of the models. In particular, we prove the following theorem, in which a learning task is parameterized by $\lambda$, a hypothesis class $\mathcal{H}$ and a class of input distributions $\mathcal{D}$ (see Section 2.1 for formal definitions).

**Theorem 1** (Main results, informal)**.** *If one-way functions exist, then for arbitrary polynomials $n < \alpha < \beta$ (e.g., $n = \lambda^{0.1}, \alpha = \lambda^5, \beta = \lambda^{10}$) over $\lambda$ the following hold.*

- ***Part 1:** There is a learning task parameterized by $\lambda$ and a robustness bound (to limit how much an adversary can perturb the inputs) such that:*

    - *The instance size is $\Theta(n)$. (That is, the length of the input is $\Theta(n)$.)*
    - *There is a robust learner that uses $\Theta(\lambda)$ parameters.*
    - *Any polynomial-time learner needs $\Theta(\alpha)$ parameters to learn the task.*

---

[7]The closeness here could mean that a human cannot tell the difference between the two images $x, x'$.

– *Any* polynomial-time *learner needs* $\Theta(\beta)$ *parameters to robustly learn the task.*

- **Part 2:** *There is a learning task parameterized by $\lambda$ and a robustness bound (to limit how much an adversary can perturb the inputs) such that:*

  – *The instance size is $\Theta(n)$.*

  – *When the adversary that generates the adversarial examples runs in polynomial time, there is an (efficient) learner that outputs a model with $O(1)$ parameters that robustly predicts the output labels with small error.*

  – *Against information-theoretic (computationally unbounded) adversaries, no learner can produce a model with $< \Theta(\alpha)$ parameters that later robustly make the predictions.*

In Sections 3 and 4, we explain the high-level ideas behind the proofs of the two parts of Theorem 1 and formalize these statements. The full proofs can be found in the supplemental material.

**Takeaway.** Here we put our work in perspective. As discussed above, prior works have considered the effect of computational efficiency (for both the learner and the attacker) on the robustness of the model. Informally, these works have shown that requiring a learner to be efficient hinders robustness, while requiring an attacker to be efficient helps achieve robustness. Our work studies the effect of computational efficiency as well but focuses on the number of parameters of the model. In spirit, we have shown a similar phenomenon. Namely, requiring a learner to be efficient increases the size of the model, while requiring an attacker to be efficient helps reduce the size of the model. Our results can be summarized as follows. In the non-robust case, requiring the learner to be efficient increases the size of the model. In the robust case: (1) Requiring the learner to be efficient increases the size of the model. This holds for robustness against both efficient and inefficient attackers. (2) Restricting to only computational efficient attackers reduces the size of the model. This holds for both efficient and inefficient learners.

**Limitations, Implications, and Open Question.** Our work shows that the phenomenon of having larger models due to computational efficiency could provably happen in certain scenarios. It does not imply, however, that this holds for all learning problems. It is a fascinating open question whether similar phenomena also happen to real-world problems. We note that this is not particular to our work, but common to most prior works in the theory of learning showing "separation" results [Bubeck et al., 2019, Degwekar et al., 2019, Mahloujifar and Mahmoody, 2019, Garg et al., 2020b].

Our results provides an explanation on why small but representative classes (e.g. 2 layers neural networks) of functions do not obtain same (robust) accuracy as larger models. This phenomenon that cannot be solely explained based on representation power of the function class might be due to computational limitations of the learning algorithm.

Finally, we note that our theorem demonstrates the separation by using the simplest setting of binary output. One can extend it to more sophisticated settings of any finite output, particularly real numbers with bounded precision. However, our work does not consider real numbers with infinite precision. In such cases, one needs to revisit computational efficiency, as the inputs are infinitely long.

## 2 Preliminaries

For a distribution $D$, $x \leftarrow D$ denotes that $x$ is sampled according to $D$. We use $U_{\mathcal{S}}$ for the uniform distribution over $\mathcal{S}$, and we use $U_n$ to represent $U_{\{0,1\}^n}$. For a set $\mathcal{S}$, $s \leftarrow \mathcal{S}$ means $s \leftarrow \mathcal{U}_{\mathcal{S}}$. For any two distributions $X$ and $Y$ over a finite universe $\Omega$, their *statistical distance* is defined as

$$\mathsf{SD}\,(X, Y) := \frac{1}{2} \cdot \sum_{\omega \in \Omega} |\mathrm{Pr}\,[X = \omega] - \mathrm{Pr}\,[Y = \omega]|\,.$$

We sometimes write $X \approx_\varepsilon Y$ to denote $\mathsf{SD}\,(X, Y) \leqslant \varepsilon$. For a vector $x = (x_1, \ldots, x_n) \in \mathbb{F}^n$ and a subset $\mathcal{S} = \{i_1, \ldots, i_\ell\} \subseteq \{1, 2, \ldots, n\}$, $x_{\mathcal{S}}$ denotes $(x_{i_1}, \ldots, x_{i_\ell})$. For any two vectors $x = (x_1, \ldots, x_n)$ and $y = (y_1, \ldots, y_n)$, their *Hamming distance* is defined as $\mathsf{HD}(x, y) = |\{i \in \{1, 2, \ldots, n\} : x_i \neq y_i\}|$. For a set $\mathcal{S}$ and an integer $0 \leqslant t \leqslant |\mathcal{S}|$, $\binom{\mathcal{S}}{t}$ represents the set of subsets of $\mathcal{S}$ of size $t$. $\mathbb{I}$ stands for the indicator function. The base for all logarithms in this paper is 2.

## 2.1 Definitions related to learning and attacks

In this subsection, we present notions and definitions that are related to learning.

We use $\mathcal{X}$ to denote the inputs and $\mathcal{Y} = \{0, 1\}$ to denote the outputs or the labels. By $\mathcal{H}$ we denote a *hypothesis class* of functions from $\mathcal{X}$ to $\mathcal{Y}$. We use $D$ to denote a distribution over $\mathcal{X} \times \mathcal{Y}$. A learning algorithm, takes a set $\mathcal{S} \in (\mathcal{X} \times \mathcal{Y})^*$ and a parameter $\lambda$ and outputs a function $f$ that is supposed to predict a fresh sample from the same distribution that has generated the set $\mathcal{S}$. The parameter $\lambda$ is supposed to capture the complexity of the problem, e.g., by allowing the inputs (and the running times) to grow. (E.g., this could be the VC dimension, but not necessarily so.) In particular, we assume that the members of the sets $\mathcal{X}_\lambda, \mathcal{Y}_\lambda$ can be represented with $\mathsf{poly}(\lambda)$ bits. A *proper* learning for a hypothesis class $\mathcal{H}$ outputs $f \in \mathcal{H}$, while an *improper* learner is allowed to output arbitrary functions. By default, we work with improper learning as we do not particularly focus on whether the learning algorithms output a function from the hypothesis class. For a set $\mathcal{S} \subseteq \mathcal{X}$ and a function $h \colon \mathcal{X} \to \mathcal{Y}$, by $\mathcal{S}^h$ we denote the *labeled* set $\{(x, h(x)) \mid x \in \mathcal{S}\}$. For a distribution $D$ and an oracle-aided algorithm $A$, by $A^D$ we denote giving $A$ access to a $D$ sampler.

**Definition 2** (Learning problems and learners). *We use $\mathcal{F} = \{F_\lambda = (\mathcal{X}_\lambda, \mathcal{Y}_\lambda, \mathcal{D}_\lambda, \mathcal{H}_\lambda)\}_{\lambda \in \mathbb{N}}$ to denote a family of learning problems where each $\mathcal{H}_\lambda$ is a set of hypothesis functions mapping $\mathcal{X}_\lambda$ to $\mathcal{Y}_\lambda$ and $\mathcal{D}_\lambda$ is a set of distributions supported on $\mathcal{X}_\lambda$.*

- *For function $\varepsilon(\cdot, \cdot)$, we say $L$ $\varepsilon$-learns $\mathcal{F}$ if*

$$\forall \lambda \in \mathbb{N}, D_\lambda \in \mathcal{D}_\lambda, h \in \mathcal{H}_\lambda, n \in \mathbb{N}; \underset{\mathcal{S} \leftarrow D_\lambda^n; f \leftarrow L(\mathcal{S}^h, \lambda)}{\mathbf{E}} [\mathrm{Risk}(h, D_\lambda, f)] \leqslant \varepsilon(\lambda, n).$$

  *where $\mathrm{Risk}(h, D_\lambda, f) = \mathrm{Pr}_{x \leftarrow D_\lambda}[h(x) \neq f(x)]$.*

- *$L$ outputs models with at most $p(\cdot)$ bits of parameters if $\forall \lambda \in \mathbb{N}$; $\left|\mathrm{Supp}\big(L(\cdot, \lambda)\big)\right| \leqslant 2^{p(\lambda)}$.*

- *$L$ is an $\varepsilon$-robust learner against (all) adversaries of budget $r$ w.r.t. distance metric $d$ if*

$$\forall \lambda \in \mathbb{N}, D_\lambda \in \mathcal{D}_\lambda, h \in \mathcal{H}_\lambda, n \in \mathbb{N}; \underset{\mathcal{S} \leftarrow D_\lambda^n; f \leftarrow L(\mathcal{S}^h, \lambda)}{\mathbf{E}} [\mathrm{Risk}_{d,r}(h, D_\lambda, f)] \leqslant \varepsilon(\lambda, n),$$

  *where $\mathrm{Risk}_{(d,r)}(h, D_\lambda, f) = \underset{x \leftarrow D_\lambda}{\mathrm{Pr}} [\max_{x'} \mathbb{I}\big(f(x') \neq h(x)\big) \wedge \mathbb{I}\big(d(x, x') \leqslant r\big)]$.*

- *$L$ runs in polynomial time if there is a polynomial $\mathsf{poly}(\cdot)$ such that for all $\lambda \in \mathbb{N}$ and $\mathcal{S} \in (\mathcal{X}_\lambda, \mathcal{Y}_\lambda)^*$, the running time of $L(\mathcal{S}, \lambda)$ is bounded by $\mathsf{poly}(|\mathcal{S}| \cdot \lambda)$.*

- *$L$ is an $\varepsilon$-robust learner against polynomial-time adversaries of budget $r$ w.r.t distance $d$, if for any family of $\mathsf{poly}(\lambda)$-time (oracle aided) adversaries $\mathcal{A} = \{A_\lambda^{(\cdot)}\}_{\lambda \in \mathbb{N}}$ we have*

$$\forall \lambda \in \mathbb{N}, D_\lambda \in \mathcal{D}_\lambda, h \in \mathcal{H}_\lambda, n \in \mathbb{N}; \underset{\mathcal{S} \leftarrow D_\lambda^n; f \leftarrow L(\mathcal{S}^h, \lambda)}{\mathbf{E}} [\mathrm{Risk}_{d,r,A_\lambda}(h, D_\lambda, f)] \leqslant \varepsilon(\lambda, n),$$

  *where $\mathrm{Risk}_{(d,r,A_\lambda)}(h, D_\lambda, f) = \underset{x \leftarrow D_\lambda, x' \leftarrow A^{D_\lambda}(x, h(x), f)}{\mathrm{Pr}} [\mathbb{I}\big(h(x') \neq f(x)\big) \wedge \mathbb{I}\big(d(x, x') \leqslant r\big)]$.*

## 2.2 Cryptographic primitives

**Definition 3** (Negligible function). *A function $\varepsilon(\lambda)$ is said to be negligible, denoted by $\varepsilon(\lambda) = \mathsf{negl}(\lambda)$, if for all polynomial $\mathsf{poly}(\lambda)$, it holds that $\varepsilon(\lambda) < 1/\mathsf{poly}(\lambda)$ for all sufficiently large $\lambda$.*

**Definition 4** (Pseudorandom generator). *Suppose $\alpha > \lambda$. A function $f \colon \{0, 1\}^\lambda \to \{0, 1\}^\alpha$ is called a pseudorandom generator (PRG) if for all polynomial-time probabilistic algorithm $A$, it holds that*

$$\left| \underset{x \leftarrow \{0,1\}^\lambda}{\mathrm{Pr}} [A(f(x)) = 1] - \underset{y \leftarrow \{0,1\}^\alpha}{\mathrm{Pr}} [A(y) = 1] \right| = \mathsf{negl}(\lambda).$$

*The ratio $\alpha/\lambda$ is refer to as the (multiplicative) stretch of the PRG.*

**Lemma 5** (Håstad et al. [1999]). *Pseudorandom generators (with arbitrary polynomial stretch) can be constructed from any one-way functions.[8]*

---

[8] A one-way function is a function that is easy to compute, but hard to invert (see Definition 21).

**Definition 6** (Signature scheme). *A signature scheme consists of three algorithms* $(\mathsf{Gen}, \mathsf{Sign}, \mathsf{Verify})$.

- $(\mathsf{vk}, \mathsf{sk}) \leftarrow \mathsf{Gen}(1^\lambda)$: *on input the security parameter* $1^\lambda$, *the randomized algorithm* $\mathsf{Gen}$ *outputs a (public) verification key* $\mathsf{vk}$ *and a (private) signing key* $\mathsf{sk}$.

- $\sigma = \mathsf{Sign}(\mathsf{sk}, m)$: *on input the signing key* $\mathsf{sk}$ *and a message* $m$, $\mathsf{Sign}$ *outputs a signature* $\sigma$.

- $b = \mathsf{Verify}(\mathsf{vk}, m, \sigma)$: *on input a verification key* $\mathsf{vk}$, *a message* $m$, *and a signature* $\sigma$, $\mathsf{Verify}$ *outputs a bit* $b$, *and* $b = 1$ *indicates tha the signature is accepted.*

*We require a (secure) signature scheme to satisfy the following properties.*

- **Correctness.** *For every message* $m$, *it holds that*

$$\Pr\left[\mathsf{Verify}(\mathsf{vk}, m, \sigma) = 1 : \begin{array}{c} (\mathsf{vk}, \mathsf{sk}) \leftarrow \mathsf{Gen}(1^\lambda) \\ \sigma = \mathsf{Sign}(\mathsf{sk}, m) \end{array}\right] = 1.$$

- **Weak unforgeability.**[9] *For any PPT algorithm* $A$ *and message* $m$, *it holds that*

$$\Pr\left[\begin{array}{c} m' \neq m \text{ and} \\ \mathsf{Verify}(\mathsf{vk}, m', \sigma') = 1 \end{array} : \begin{array}{c} (\mathsf{vk}, \mathsf{sk}) \leftarrow \mathsf{Gen}(1^\lambda) \\ \sigma = \mathsf{Sign}(\mathsf{sk}, m) \\ (m', \sigma') \leftarrow A(\mathsf{vk}, m, \sigma) \end{array}\right] = \mathsf{negl}(\lambda).$$

**Lemma 7** (Naor and Yung [1989], Rompel [1990])**.** *Signature schemes can be constructed from any one-way function.*

## 2.3 Coding theory

Let $\mathbb{F}$ be a finite field. A *linear code* $\mathcal{H}$ with *block length* $n$ and *dimension* $k$ is a $k$-dimensional subspace of the vector space $\mathbb{F}^n$. The *generator matrix* $G \in \mathbb{F}^{k \times n}$ maps every message $m \in \mathbb{F}^k$ to its encoding, namely, the message $m$ is encoded as $m \cdot G$. The *distance* $d$ of the code $\mathcal{H}$ is the minimum distance between any two codewords. That is, $d = \min_{(x,y \in \mathcal{H}) \wedge (x \neq y)} \mathsf{HD}(x, y)$. The *rate* of the codeword is defined as $R = k/n$.

**Reed-Solomon code.** In this work, we will mainly use the Reed-Solomon (RS) code. For the RS code, every message $m = (m_1, \ldots, m_k) \in \mathbb{F}^k$ is parsed as a degree $k - 1$ polynomial

$$f(x) = m_1 + m_2 \cdot x + \cdots + m_k \cdot x^{k-1}$$

and the encoding is simply $(f(1), f(2), \ldots, f(n)) \in \mathbb{F}^n$.

**Definition 8** (List-decodable code)**.** *A code* $\mathcal{H} \subseteq \mathbb{F}^n$ *is said to be* $(p, L)$-*list-decodable if for any string* $x \in \mathbb{F}^n$, *there are* $\leqslant L$ *messages whose encoding* $\mathsf{c}$ *satisfies* $\mathsf{HD}(x, \mathsf{c}) \leqslant p \cdot n$. *We say the code* $\mathcal{H}$ *is efficiently* $(p, L)$-*list-decodable if there is an efficient algorithm that finds all such messages.*

**Lemma 9** (Guruswami and Sudan [1998])**.** *For any constant* $R > 0$, *the Reed-Solomon code with constant rate* $R$ *and block length* $n$ *is efficiently* $(1 - \sqrt{R}, \mathsf{poly}(n))$-*list-decodable.*

## 2.4 Randomness extraction

**Definition 10** (Min-entropy)**.** *The min-entropy of a distribution* $X$ *over a finite set* $\Omega$ *is defined as*

$$H_\infty(X) := -\log\left(\max_{\omega \in \Omega} \Pr[X = \omega]\right).$$

We need the following result about the sampler. A sampler (for a target set of coordinates $\{1, 2, \ldots, n\}$) is a deterministic mapping that only takes randomness as input and outputs a subset of $\{1, 2, \ldots, n\}$. Below, we let $\binom{[n]}{t} = \{\mathcal{S} \mid |\mathcal{S}| = t, \mathcal{S} \subseteq [n]\}$.

---

[9]We consider a rather weak notion of unforgability. Here, we require that the adversary cannot forge a signature when he is given only one valid pair of message and signature. This weaker security notion already suffices for our purposes. In the stronger notion, the adversary is allowed to pick $m$ based on the given $\mathsf{vk}$.

**Lemma 11** (Lemma 6.2 and Lemma 8.4 of Vadhan [2004]). *For any $0 < \kappa_1, \kappa_2 < 1$ and any $n, t \in \mathbb{N}$, there exists a deterministic sampler $\mathsf{samp} \colon \{0,1\}^r \to \binom{[n]}{t}$ such that the following hold.*

- *If $X$ is a random variable over $\{0,1\}^n$ with min-entropy $\geqslant \mu \cdot n$, there exists a random variable $Y$ over $\{0,1\}^t$ with min-entropy $\geqslant (\mu - \kappa_1) \cdot t$ and*

$$\mathsf{SD}\left(\,\left(U_r, X_{\mathsf{samp}(U_r)}\right),\, \left(U_r, Y\right)\,\right) \leqslant \exp(-\Theta(n\kappa_1)) + \exp\left(-n^{\kappa_2}\right),$$

  *where the two $U_r$ in the first joint distribution refer to the same sample.*

- *Furthermore, $r = \Theta\left(n^{\kappa_2}\right)$.*

This lemma by Vadhan Vadhan [2004] states that one could use a small amount of randomness to sub-sample from a distribution $X$ with the guarantee that $X_{\mathsf{samp}(U_r)}$ is close to another distribution $Y$ that preserves the same min-entropy rate as $X$.

Due to space limitations, additional preliminaries are included in Appendix A.

## 3 Efficient learner vs. information-theoretic learner

In this section, we explain the ideas behind the proof of Part 1 of Theorem 1. We start with the high-level ideas behind our construction. Then, we formally state the construction of the learning task and its properties through formal statements. The proofs can be found in the supplemental material.

Consider the problem of learning an inner product function $\mathsf{IP}_P$ defined as $\mathsf{IP}_P(x) = \langle x, P \rangle$, where the inner product is done in $\mathbb{GF}_2$. Our first observation is that to learn $\mathsf{IP}_P$ with a small error where $P$ is uniformly random, the number of model parameters that the learner employs must be as (almost) as large as the length of $P$. Intuitively, one can argue it as follows. Let $Z$ denote the parameters in the model that the learner outputs. Suppose that $Z$ is shorter than $P$. Then $P$ must still be unpredictable given $Z$.[10] By a standard result in randomness extraction, one can argue that, for a uniform $x$,

$$(Z, x, \langle x, P \rangle) \approx (Z, x, U_{\{0,1\}}).$$

That is, the label $\langle x, P \rangle$ looks information-theoretically uniform to the classifier who holds $Z$ and $x$. Therefore, the classifier can only output the correct label with (the trivial) probability $\approx 1/2$.

The conclusion is that learning *all* linear functions need a learner that outputs as many parameters as the function's description is. However, even an *efficient* learner can perform the learning just as well as an information-theoretic one (e.g., using the Gaussian elimination). We now show how to modify this task to make a big difference between an efficient learner and an information-theoretic learner in terms of the number of parameters that they output.

**Computational perspective.** Now, suppose $P$ is *computationally pseudorandom* [Goldwasser and Micali, 1984] rather than being truly random.[11] That is, let $f : \{0,1\}^\lambda \to \{0,1\}^\alpha$ be a pseudorandom generator (Definition 4) and $P$ is distributed as $f(U_\lambda)$, where $U_\lambda$ denotes the uniform distribution over $\lambda$ bits. Since (1) an efficient learner cannot distinguish $P \leftarrow f(U_\lambda)$ from $P \leftarrow U_\alpha$ and (2) any learner who tries to learn $\mathsf{IP}_P$ with $P \leftarrow U_\alpha$ needs $\alpha$ parameters, it can be proved that an efficient learner who tries to learn $\mathsf{IP}_P$ with $P \leftarrow f(U_\lambda)$ must also uses $\alpha$ parameters. However, an information-theoretic learner needs only $\lambda$ parameters to learn $\mathsf{IP}_P$ with $P \leftarrow f(U_\lambda)$ as it can essentially find the seed $s$ such that $P = f(s)$ and output the seed $s$. To conclude, to learn $\mathsf{IP}_P$ for $P \leftarrow f(U_\lambda)$, an information-theoretic learner only needs a few (i.e., $\lambda$) parameters and an efficient learner needs a lot of (i.e., $\alpha$) parameters. We emphasize that for a pseudorandom generator, its output length $\alpha$ could have an arbitrarily large polynomial dependence on its input length $\lambda$. For instance, it could be $\alpha = \lambda^{10}$.

**Robustness.** We now explain the ideas behind Theorem 15 in which we study the role of model robustness in the size of the model parameters. We now suppose the instance is sampled according to the distribution $D_Q$ (parametrized by a string $Q$), which is defined by

$$\mathsf{Enc}(U) + Q.$$

---

[10]That is, with high probability over the choice of $Z = z$, the conditional distribution $P|(Z = z)$ has high min-entropy. More formally, this unpredictability is measured by the average-case min-entropy (Definition 23).

[11]A pseudorandom string is indistinguishable from random ones for computationally bounded distinguishers.

Here, $U$ is the uniform distribution (over the right number of bits), $\mathsf{Enc}(U)$ is an error-correcting encoding of $U$, and the addition is coordinate-wise field addition. Moreover, the label for this instance is the inner product $\langle U, P \rangle$ for some vector $P$. Observe that if the learner learns $Q$, it can always find the correct label on a perturbed instance due to the error-correcting property of $\mathsf{Enc}(U)$. We argue that, in order to robustly learn this task for a uniformly random $Q$, the number of parameters in the model that the learner outputs must be almost as large as the length of $Q$. Intuitively, the argument is as follows. Let $Z$ be the model that the learner outputs. Since $|Z| < |Q|$, then $Q$ must still be unpredictable given $Z$. In this case, we prove that

$$(Z, \mathsf{Enc}(U) + Q + \rho) \approx (Z, U').$$

Here, $\rho$ stands for the noise that the adversary adds to the instance. In words, the classifier holding the parameter $Z$ cannot distinguish the perturbed instance $\mathsf{Enc}(U) + Q + \rho$ from a uniformly random string $U'$. Since $U$ is information-theoretically hidden to the classifier, it can only output the correct label $\langle U, P \rangle$ with probability $\approx 1/2$.

Next, to explore the difference between an efficient and information-theoretic learner, we consider the case where $Q$ is pseudorandomly distributed, i.e. $Q \leftarrow f(U_\lambda)$ for some $f : \{0,1\}^\lambda \to \{0,1\}^\beta$. Again, since (1) an efficient learner cannot distinguish $Q \leftarrow U_\beta$ from $Q \leftarrow f(U_\lambda)$ and (2) any learner needs at least $\approx |Q|$ parameters to learn the task with $Q \leftarrow U_\beta$, an efficient learner must also need at least $\approx |Q|$ parameters to learn the task. On the other hand, an information-theoretic learner could again find the seed $s$ such that $Q = f(s)$ and output the seed $s$ as the parameter. To conclude, an information-theoretic learner requires few (i.e., $\lambda$) parameters to robustly learn the task and an efficient learner needs a lot of (i.e., $\approx \beta$) parameters to robustly learn the task. (Again, $\beta$ could have an arbitrary polynomial dependence on $\lambda$.)

**Making instances small.** The learning tasks we considered above suffer from one drawback: the size of the instance is very large, or at least is related to the number of parameters of the model. Here we ask: is it possible to have a small instance size while an efficient learner still needs to output a very large model? We answer this question positively. In particular, for any $n = \mathsf{poly}(\lambda)$ (e.g., $n = \lambda^{0.1}$), we construct a learning problem where the instance size is $\Theta(n)$ and the efficient learner still needs $\alpha$ (resp. $\beta$) parameters to learn (resp. robustly learn) the task. Our construction uses the "sampler" by Vadhan [2003]. Informally, a sampler $\mathsf{samp}$ (see Lemma 11) needs a small seed $u$ and outputs a subset of $\{1, 2, \ldots, \alpha\}$ of size $n$. The sampler comes with the guarantee that if $P$ is a source with high entropy, $P|_{\mathsf{samp}(u)}$ also has high enough entropy. To illustrate the usage of the sampler, consider learning this new inner product function $\mathsf{IP}_P$ defined as

$$\mathsf{IP}_P(u, x) = \left\langle x, P|_{\mathsf{samp}(u)} \right\rangle.$$

For uniformly random $P$, one can similarly argue that a model must output at least $|P|$ parameters to learn the task. Let $Z$ denote the parameters in the model that the learner outputs. If $|Z| < |P|$, then $P$ contains high entropy conditioned on $Z$. By the property of the sampler, it must hold that $P|_{\mathsf{samp}(u)}$ contains high entropy conditioned on $Z$ and $u$. Consequently,

$$(Z, u, x, \mathsf{IP}_P(u, x)) \approx (Z, u, x, U_{\{0,1\}}).$$

Namely, the classifier who sees the parameter $Z$ and the instance $(u, x)$ can only predict the label with probability $\approx 1/2$. Observe that the size of the instance $(u, x)$ is roughly $n$,[12] while $P$ could have length $\alpha \gg n$. The use of the sampler in the robust learning case is similar to the non-robust case above. We refer the readers to Appendix B for more details.

To sum up, we present our formal construction and formal theorem statement below. The formal proof is presented in the supplemental material.

**Construction 12** (Parameter-heavy models under efficient learning)**.** *Given the parameter $n < \lambda < \alpha < \beta$, we construct the following learning problem.[13] We rely on the following building blocks.*

- *Let $f_1 : \{0,1\}^\lambda \to \{0,1\}^\alpha$ and $f_2 : \{0,1\}^\lambda \to \{0,1\}^\beta$ be PRGs (Definition 4).*

- *Let $\mathsf{Enc}$ be a RS encoding with dimension $k$, block length $n$, and rate $R = k/n$. The rate is chosen to be any constant $< 1/3$ and $k$ is defined by $R$ and $n$. This RS code is over the field $\mathbb{F}_{2^\ell}$ for some $\ell = \Theta(\log \lambda)$.*

---

[12]As the seed $u$ is very small.

[13]All the other parameters are implicitly defined by these parameters.

- *Let* $\mathsf{samp}_1 \colon \{0,1\}^{r_1} \to \binom{\{1,\ldots,\alpha\}}{k\cdot\ell}$ *and* $\mathsf{samp}_2 \colon \{0,1\}^{r_2} \to \binom{\{1,\ldots,\beta\}}{n\cdot\ell}$ *be samplers. We obtain these samplers by invoking Lemma 11 with sufficiently small $\kappa_1$ and $\kappa_2$ (e.g., $\kappa_1 = \Theta(1/\log\lambda)$ and sufficiently small constant $\kappa_2$). Note that $\kappa_1, \kappa_2$ define $r_1, r_2$.*

- *For any binary string $v$, we use $[v]$ for an arbitrary error-correcting encoding of $v$ (over the field $\mathbb{F}_{2^\ell}$) such that $[v]$ can correct $> (1-R)n/2$ errors. This can always be done by encoding $v$ using RS code with a suitable (depending on the dimension of $v$) rate. Looking forward, we shall consider an adversary that may perturb $\leqslant (1-R)n/2$ symbols. Therefore, when a string $v$ is encoded as $[v]$ and the adversary perturb it to be $\widetilde{[v]}$, it will always be error-corrected and decoded back to $v$.*

*We now construct the following learning task $F_\lambda = (\mathcal{X}_\lambda, \mathcal{Y}_\lambda, \mathcal{D}_\lambda, \mathcal{H}_\lambda)$.*

- $\mathcal{X}_\lambda$ *implicitly defined by the distribution $\mathcal{D}_\lambda$, and $\mathcal{Y}_\lambda = \{0,1\}$.*

- $\mathcal{D}_\lambda$ *consists of distributions $D_s$ for $s \in \{0,1\}^\lambda$, where $D_s$ is*

$$D_s = \Big([u_1],\ [u_2],\ m,\ \mathsf{Enc}(m) + \Big(f_2(s)\big|_{\mathsf{samp}_2(u_2)}\Big)\Big),\ \text{where}$$

  - $u_1$ *and $u_2$ are sampled uniformly at random from $\{0,1\}^{r_1}$ and $\{0,1\}^{r_2}$, respectively.*
  - $m$ *is sampled uniformly at random from $\mathbb{F}_{2^\ell}^k$.*
  - $f_2(s)\big|_{\mathsf{samp}_2(u_2)}$ *is interpet as a vector over $\mathbb{F}_{2^\ell}$ and $\mathsf{Enc}(m) + \Big(f_2(s)\big|_{\mathsf{samp}_2(u_2)}\Big)$ is coordinate-wise addition over $\mathbb{F}_{2^\ell}$.*

- $\mathcal{H}_\lambda$ *consists of all functions $h_s \colon \mathcal{X} \to \mathcal{Y} = \{0,1\}$ for all $s \in \{0,1\}^\lambda$, where $h_s$ is:*

$$h_s(x) = \Big\langle m, \Big(f_1(s)\big|_{\mathsf{samp}_1(u_1)}\Big)\Big\rangle,$$

  *and the inner product is over $\mathbb{F}_2$ and $m$ is interpreted as a string $\in \mathbb{F}_2^{k\cdot\ell}$ in the natural way.*

- ***Adversary.*** *The entire input $x = ([u_1], [u_2], m, \mathsf{Enc}(m) + (f_2(s)|_{\mathsf{samp}_1(u_1)}))$ is interpreted as a vector over $\mathbb{F}_{2^\ell}$ and we consider an adversary that may perturb $\leqslant (1-R)n/2$ symbols. That is, the adversary has a budget of $(1-R)n/2$ in Hamming distance over $\mathbb{F}_{2^\ell}$.*

**Theorem 13.** *An information-theoretic learner can (robustly) $\varepsilon$-learn the task of Construction 12 with parameter size $2\lambda$ and sample complexity $\Theta(\frac{\lambda}{\varepsilon})$. Moreover, an efficient learner can (robustly) $\varepsilon$-learn this task with parameter size $\alpha + \beta$ and sample complexity $\Theta(\frac{\alpha+\beta}{\varepsilon})$.*

**Theorem 14.** *Any efficient learner that outputs models with $\leqslant \alpha/2$ parameters cannot $\varepsilon$-learn $F_\lambda$ of Construction 12 for $\varepsilon < 1/3$.*

**Theorem 15.** *There exists some constant $c$ such that the following holds. In the presence of an adversary that may perturb $(1-R)n/2$ symbols, any efficient learner for the task of Construction 12 that outputs a model with $c \cdot \beta/\log\lambda$ parameters cannot $\varepsilon$-robustly learn $F_\lambda$ for $\varepsilon < 1/3$.*

## 4 Efficient adversary vs. information-theoretic adversary

In this section, we explain the ideas behind the proof of Part 2 of Theorem 1. At the end of this section, we formally state the construction of the learning task and state its properties through formal statements. The proofs can be found in the supplemental material.

We now explore whether the computational efficiency of the adversary could affect the number of parameters required to robustly learn a task. Garg et al. [2020a] considered the difference between an efficient adversary and an information-theoretic one in terms of their running time. We first recall their construction. The learning instance is sampled from the distribution

$$[\mathsf{vk}], b, \mathsf{Sign}(\mathsf{sk}, b).$$

Here, $(\mathsf{vk}, \mathsf{sk})$ is the verification key and signing key pair of a signature scheme (Definition 6);[14] $[\mathsf{vk}]$ is an error-correcting encoding of $\mathsf{vk}$, which ensures that $[\mathsf{vk}]$ can always be recovered after perturbation by the adversary; $b$ is a uniform random bit and the label of the instance is simply $b$.

---

[14]Every instance samples a fresh pair of verification key and signing key $(\mathsf{vk}, \mathsf{sk})$.

The signature scheme ensures that an efficient adversary cannot forge a valid signature and, hence, any efficient adversary that perturbs $(b, \mathsf{Sign}(\mathsf{sk}, b))$ will result in an invalid message/signature pair. A classifier could then detect such perturbations and output a special symbol $\perp$ indicating that perturbation is detected. On the other hand, an information-theoretic adversary could launch a successful attack by forging a valid signature $(1 - b, \mathsf{Sign}(\mathsf{sk}, 1 - b))$. Therefore, it could perturb the instance to be $[\mathsf{vk}], 1 - b, \mathsf{Sign}(\mathsf{sk}, 1 - b)$ and, hence, flipping the label of the output.

**First idea.** We want to construct a learning problem such that the learner needs few parameters against an efficient adversary, but a lot of parameters against an information-theoretic adversary. Our first idea is to add another way of recovering $b$ in the above learning problem. Consider the learning problem where the instance is sampled from distribution $D_P$ defined as

$$[\mathsf{vk}], b, \mathsf{Sign}(\mathsf{sk}, b), [b + \langle u, P \rangle], [u].$$

Here, $u$ is uniformly distributed. Observe that if the learner learns $P$, it can recover $b$ correctly from $[b + \langle u, P \rangle]$ and $[u]$ by error-correction decoding. However, if the number of parameters in the model that the learner outputs have $< |P|$ parameters, then $[b + \langle u, P \rangle]$ and $[u]$ information-theoretically hides $b$. Again, this is because $P$ is unpredictable given the parameter $Z$[15] and, hence,

$$Z, u, \langle u, P \rangle \approx Z, u, U_{\{0,1\}}.$$

Consequently, an information-theoretic adversary could again launch the attack that replace $b, \mathsf{Sign}(\mathsf{sk}, b)$ with $1 - b, \mathsf{Sign}(\mathsf{sk}, 1 - b)$ and successfully flipping the label. Therefore, a learner must employ $|P|$ parameters to robustly learn the task against information-theoretic adversaries.

**Second idea.** In the above learning task, a learner employing $|P|$ parameters can always recover the correct label against information-theoretic adversaries. However, against efficient adversaries, a learner with fewer parameters may not always recover the correct label but will sometimes output the special symbol $\perp$ indicating that tampering is detected. Could we twist it to ensure that a learner with fewer parameters can also always recover the correct label against efficient adversaries? We positively answer this question by using list-decodable code (Definition 8). Intuitively, list-decoding ensures that given an erroneous codeword, the decoding algorithm will find the list of all messages whose encoding is close enough to the erroneous codeword.

Our new learning task has instances drawing from the distribution $D_P$ defined as

$$[\mathsf{vk}], \mathsf{LEnc}(b, \mathsf{Sign}(\mathsf{sk}, b)), [b + \langle u, P \rangle], [u].$$

Here, $\mathsf{LEnc}$ is the encoding algorithm of the list-decoding code. The main idea is that: after the perturbation on $\mathsf{LEnc}(b, \mathsf{Sign}(\mathsf{sk}, b))$, the original message/signature pair $b, \mathsf{Sign}(\mathsf{sk}, b)$ will always appear in the list output by the decoding algorithm. Now, against an efficient adversary, $b, \mathsf{Sign}(\mathsf{sk}, b)$ must be the only valid message/signature pair in the list. Otherwise, this adversary breaks the unforgeability of the signature scheme. Against an information-theoretic adversary, however, it could introduce $(1 - b, \mathsf{Sign}(\mathsf{sk}, 1 - b))$ into the list recovered by the list-decoding algorithm. Consequently, the learner cannot tell if the correct label is $b$ or $1 - b$. Consequently, for this learning task, against information-theoretic adversaries, one still needs $|P|$ parameters. Against efficient adversaries, one needs only a few parameters and can always recover the correct label.

**Making instances small.** Again, we have this unsatisfying feature that the instance has the same size as the model. We resolve this issue using the sampler in a similar way. We refer the readers to Appendix C for more details.

We present our formal construction and theorem statement below. The formal proof is in Appendix C.

**Construction 16** (Learning task for bounded/unbounded attackers). *Given the parameters $n < \lambda < \alpha$, we construct the following learning problem.[16] We use the following tools.*

- (Gen, Sign, Verify) *be a signature scheme (see Definition 6).*

- *Let* $\mathsf{LEnc}$ *be a RS encoding with dimension $k$, block length $n$, and rate $R = k/n$. We pick the rate $R$ to be any constant $< 1/4$ and $k$ is defined by $R$ and $n$. This RS code is over the field $\mathbb{F}_{2^\ell}$ for some $\ell = \Theta(\log \lambda)$.*

---

[15]That is, with high probability over the choice of $Z = z$, the conditional distribution $P|(Z = z)$ has high min-entropy. More formally, this unpredictability is measured by the average-case min-entropy (Definition 23).

[16]All the other parameters are implicitly defined by these parameters.

- *Let* samp$\colon \{0,1\}^r \to \binom{\{1,\dots,\alpha\}}{n}$ *be samplers. (We obtain these samplers by invoking Lemma 11 with sufficiently small $\kappa_1$ and $\kappa_2$. For instance, setting $\kappa_1 = \Theta(1/\log \lambda)$ and $\kappa_2$ to be any small constant suffices.)*

- *For any binary string $v$, we use $[v]$ for an arbitrary error-correcting encoding of $v$ (over the field $\mathbb{F}_{2^\ell}$) such that $[v]$ can correct $> (1 - \sqrt{R})n$ errors. This can always be done by encoding $v$ using RS code with a suitable (depending on the dimension of $v$) rate. Looking forward, we shall consider an adversary that may perturb $\leqslant (1 - \sqrt{R})n$ symbols. Therefore, when a string $v$ is encoded as $[v]$ and the adversary perturbs it to be $\widetilde{[v]}$, it will always be error-corrected and decoded back to $v$.*

*We now construct the following learning task $F_\lambda = (\mathcal{X}_\lambda, \mathcal{Y}_\lambda, \mathcal{D}_\lambda, \mathcal{H}_\lambda)$.*

- *$\mathcal{X}_\lambda$ is $\{0,1\}^N$ for some $N$ that is implicitly defined by $\mathcal{D}_\lambda$, and $\mathcal{Y}_\lambda$ is $\{0,1\}$.*

- *The distribution $\mathcal{D}_\lambda$ consists of all distribution $D_s$ for $s \in \{0,1\}^\alpha$*

$$D_s = \Big( [u], [v], \ [\mathsf{vk}], \ \mathsf{LEnc}(b, \mathsf{Sign}(\mathsf{sk}, b)), \ \big[b + \langle v, s|_{\mathsf{samp}(u)} \rangle \big] \Big), \ \textit{such that}$$

  - *$u$ are sampled uniformly from $\{0,1\}^r$ and $v$ is sampled uniformly from $\{0,1\}^n$.*
  - *$(\mathsf{vk}, \mathsf{sk}) \leftarrow \mathsf{Gen}(1^\lambda)$ are sampled from the signature scheme.*
  - *$b$ is sampled uniformly at random from $\{0,1\}$. $(b, \mathsf{Sign}(\mathsf{sk}, b))$ is intepreted as a vector in $\mathbb{F}_{2^\ell}^k$ in the natural way.*

- *$h_\lambda$ consists of one single function $h$. On input $x = ([u], [v], \mathsf{LEnc}(b, \mathsf{Sign}(\mathsf{sk}, b), [b + \langle v, s|_{\mathsf{samp}(u)} \rangle])$, $h(x)$ simply decodes $\mathsf{LEnc}(b, \mathsf{Sign}(\mathsf{sk}, b))$ and output $b$.*

- ***Adversary.** The entire input $([u], [v], [\mathsf{vk}], \mathsf{LEnc}(b, \mathsf{Sign}(\mathsf{sk}, b)), [b + \langle v, s|_{\mathsf{samp}(u)} \rangle])$ is interpreted as a vector over $\mathbb{F}_{2^\ell}$ and we consider an adversary that may perturb $\leqslant (1 - \sqrt{R})n$ symbols. That is, the adversary has a budget of $(1 - \sqrt{R})n$ for Hamming distance over $\mathbb{F}_{2^\ell}$.*

**Theorem 17.** *For the learning task of Construction 16, there is an efficient learner (with 0 sample complexity) that outputs a model with no parameter and $\mathsf{negl}(\lambda)$-robustly learns $F_\lambda$ against computationally-bounded adversaries of budget $(1 - \sqrt{R})n$.*

**Theorem 18.** *For computationally unbounded adversaries, any information-theoretic learner with $\alpha/2$ parameters cannot $\varepsilon$-robustly learn $F_\lambda$ for $\varepsilon < 1/3$ for the learning task of Construction 16.*

## 5  Acknowledgement

Sanjam Garg and Mingyuan Wang are supported by DARPA under Agreement No. HR00112020026, AFOSR Award FA9550-19-1-0200, NSF CNS Award 1936826, and research grants by the Sloan Foundation, and Visa Inc. Any opinions, findings and conclusions or recommendations expressed in this material are those of the author(s) and do not necessarily reflect the views of the United States Government or DARPA. Mohammad Mahmoody is supported by NSF grants CCF-1910681 and CNS1936799. Somesh Jha is partially supported by Air Force Grant FA9550-18-1-0166, the National Science Foundation (NSF) Grants CCF-FMitF-1836978, IIS-2008559, SaTC-Frontiers-1804648, CCF-2046710 and CCF-1652140, and ARO grant number W911NF-17-1-0405. Somesh Jha is also partially supported by the DARPA-GARD problem under agreement number 885000.

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
