# A    Supplementary material

## A.1    Learning

Our proof also relies on the following theorem from Bubeck et al. [2019].

**Theorem 19** (Implied by Theorem 3.1 of Bubeck et al. [2019])**.** *Let $\{\mathcal{C}_\lambda\}_\lambda$ be a finite family of classifiers. Suppose the learning problem $\mathcal{F} = \{\mathcal{F}_\lambda\}_\lambda$ and a learner $L$ satisfy the following. For all $D_\lambda \in \mathcal{D}_\lambda$ and $h \in \mathcal{H}_\lambda$, and sample $\mathcal{S} \leftarrow D_\lambda^n$, $L(\mathcal{S}^h)$ always outputs a classifier $f \in \mathcal{C}_\lambda$ such that $\mathrm{Risk}_{d,r}(h, D_\lambda, f) = 0$ (i.e., $f$ robustly fits $h$ perfectly). Then, $L$ will $\delta$-robust learn $\mathcal{F}$ with sample complexity $\log|\mathcal{C}_\lambda|/\delta$.*

We emphasize that in the theorem above, one might pick a *different* set of classifiers merely for sake of computational efficiency of the learner $L$. Namely, it might be possible to information-theoretically learn a hypothesis class robustly (e.g., by a robust variant of empirical risk minimization when), but an efficient learner might choose to output its classifiers from a larger set such that it can *efficiently* find a member of that class.

## A.2    Cryptographic primitives

**Definition 20** (Computational indistinguishability)**.** *We say two ensembles of distributions $X = \{X_\lambda\}_{\lambda \in \mathbb{N}}$ and $Y = \{Y_\lambda\}_{\lambda \in \mathbb{N}}$ are computationally indistinguishable if for any probabilistic polynomial-time (PPT) algorithm $A$, it holds that*

$$\left| \Pr_{x \leftarrow X_\lambda} [A(x) = 1] - \Pr_{y \leftarrow Y_\lambda} [A(y) = 1] \right| = \mathsf{negl}(\lambda).$$

**Definition 21** (One-way function)**.** *An ensemble of functions $\{f_\lambda : \{0,1\}^\lambda \to \{0,1\}^\lambda\}_\lambda$ is called a one-way function if for all polynomial-time probabilistic algorithm $A$, it holds that*

$$\Pr \begin{bmatrix} x \leftarrow \{0,1\}^\lambda,\ y = f_\lambda(x) \\ x' \leftarrow A(1^\lambda, y) \end{bmatrix} :\ f_\lambda(x') = y \end{bmatrix} = \mathsf{negl}(\lambda).$$

## A.3    Coding theory

**Fact 22.** *The following facts hold about the Reed-Solomon code.*

- *The distance of the Reed-Solomon code is $d = n - k + 1$. Moreover, the decoding is possible efficiently: there is a PPT algorithm that maps any erroneous codeword that contains up to $\leqslant (n-k)/2$ errors to the nearest correct (unique) codeword. In other words, one can efficiently correct up to $\frac{1-R}{2}$ fraction of errors, where $R$ is the code's rate.*

- *The encoding of a* random *message is $k$-wise independent. That is, for all subset $\mathcal{S} \subseteq \{1, 2, \ldots, n\}$ such that $|\mathcal{S}| \leqslant k$, the following distribution*

$$\left\{ \begin{matrix} m \leftarrow \mathbb{F}^k,\ \mathsf{c} = m \cdot G \\ \text{Output } \mathsf{c}_\mathcal{S} \end{matrix} \right\}$$

*is uniform over $\mathbb{F}^{|\mathcal{S}|}$. This follows from the fact that any $\leqslant k$ columns of the generator matrix of the RS code is full-rank.*

## A.4    Randomness Extraction

In certain cases, some information regarding $X$ is learned (e.g., through training). Let us denote this learned information as a random variable $Z$. Note that $X$ and $Z$ are two correlated distributions. In order to denote the min-entropy of $X$ conditioned on the learned information $Z$, we need the following notion (and lemma) introduced by Dodis et. al. Dodis et al. [2008].

**Definition 23** (Average-case min-entropy Dodis et al. [2008])**.** *For two correlated distributions $X$ and $Z$, the average-case min-entropy is defined as*

$$\widetilde{H}_\infty(X|Z) = -\log \left( \mathop{\mathrm{E}}_{z \leftarrow Z} \left[ \max_x \Pr[X = x | Z = z] \right] \right).$$

**Lemma 24** (Dodis et al. [2008]). *We have the following two lemmas regarding the average-case min-entropy.*

- *If the support set of $Z$ has size at most $2^m$, we have*

$$\widetilde{H}_\infty(X|Z) \geqslant H_\infty(X) - m.$$

- *It holds that*

$$\Pr_{z \leftarrow Z}\left[H_\infty(X|Z=z) \geqslant \widetilde{H}_\infty(X|Z) - \log(1/\varepsilon)\right] \geqslant 1 - \varepsilon.$$

Intuitively, the above lemma states that: if a model denoted by the random variable $Z$ is not too large, and that model captures the information revealed about a variable $X$, since the support set of $Z$ has small size, then the average-case min-entropy $\widetilde{H}_\infty(X|Z)$ is large. Furthermore, for most $z$, the min-entropy of $H_\infty(X|Z=z)$ is almost as large as $\widetilde{H}_\infty(X|Z)$.

We now recall a tool that, roughly speaking, states that if $X$ is a distribution over $\{0,1\}^n$ that contains some min-entropy, then the inner product (over $\mathbb{F}_2$) between $X$ and a random vector $Y$ is a uniformly random bit, even conditioned on most of $Y$. This is a special case of the celebrated *leftover hash lemma* Håstad et al. [1999]. We summarize this result as the following theorem. For completeness, a proof can be found in Appendix D.1.

**Theorem 25** (Inner product is a good randomness extractor). *For all distribution $X$ over $\{0,1\}^n$ such that $H_\infty(X) \geqslant 2 \cdot \log(1/\varepsilon)$, it holds that*

$$(U_n, \langle X, U_n \rangle) \approx_\varepsilon U_{n+1},$$

*where the two $U_n$ refer to the same sample.*

Next, we need the following notion and results from Fourier analysis.

**Definition 26** (Small-bias distribution Naor and Naor [1993]). *Let $\mathbb{F}_{2^\ell}$ be the finite field of order $2^\ell$. For a distribution $X$ over $\mathbb{F}_{2^\ell}^n$, the bias of $X$ with respect to a vector $y \in \mathbb{F}_{2^\ell}^n$ is defined as*

$$\mathsf{bias}(X, \alpha) := \left| \mathop{\mathrm{E}}_{x \leftarrow X}\left[ (-1)^{\mathsf{Tr}(\langle x, \alpha \rangle)} \right] \right|,$$

*where $\mathsf{Tr}\colon \mathbb{F}_{2^\ell} \to \mathbb{F}_2$ denote the* trace map *defined as $\mathsf{Tr}(y) = y + y^{2^1} + y^{2^2} + \cdots + y^{2^{\ell-1}}$. The distribution $X$ is said to be $\varepsilon$-small-biased if for all non-zero vector $\alpha \in \{0,1\}^n$, it holds that*

$$\mathsf{bias}(X, \alpha) \leqslant \varepsilon.$$

Note that the trace map maps elements from $\mathbb{F}_{2^\ell}$ to $\mathbb{F}_2$, where exactly half of the field elements maps to 1 and the other half to 0. Consequently, if $\langle X, \alpha \rangle$ is a uniform distribution over $\mathbb{F}_{2^\ell}$, then $\mathsf{bias}(X, \alpha) = 0$. In Appendix D.2, prove the theorem below.

**Theorem 27** (Small-biased Masking Lemma Dodis and Smith [2005]). *Let $X$ and $Y$ be distributions over $\mathbb{F}_{2^\ell}^n$. If $H_\infty(X) \geqslant k$ and $Y$ is $\varepsilon$-small-biased, it holds that*

$$\mathsf{SD}\left(X + Y, U_{\{0,1\}^{n\ell}}\right) \leqslant 2^{\frac{n\ell-k}{2}-1} \cdot \varepsilon.$$

Finally, we observe the following property about the noisy RS code. A proof can be found in Appendix D.3.

**Theorem 28** (Noisy RS code is small-biased). *Let $\mathcal{H}$ be a RS code over $\mathbb{F}_{2^\ell}$ with block length $n$ and rate $R$. For all integer $s \leqslant n$, consider the following distribution*

$$D = \left\{ \begin{array}{c} \mathsf{c} \leftarrow \mathcal{H} \\ \textit{Sample a random } \mathcal{S} \subseteq \{1, 2, \ldots, n\} \textit{ such that } |\mathcal{S}| = s \\ \forall i \in \mathcal{S}, \textit{ replace } \mathsf{c}_i \textit{ with a random field element} \\ \textit{Output } \mathsf{c} \end{array} \right\}.$$

*It holds that $D$ is $(1-R)^s$-small-biased.*

## B Efficient learning could need more model parameters

In this section, we formally prove the first part of Theorem 1, which is the separation result between the number of parameters needed by unbounded v.s. bounded learners.

Our construction of the learning problem in presented in Construction 12.

First, observe that instance size is approximately $\Theta((k+n) \cdot \ell) = \Theta(n \cdot \log \lambda)$ as the size of the sampler inputs $u_1$ and $u_2$ are sufficiently small. Our theorems prove the following. First, in Theorem 13 we establish the efficient (robust) learnability of the task of Construction 12, where the efficient-learner variant requires more parameters. Then, in Theorem 14 we establish the lower bound on the number of parameters needed by an efficient learner. Finally, in Theorem 15 we establish the lower bound on the number of parameters needed by efficient *robust* learners.

In the rest of this section, we prove these theorems.

**Theorem** (Restatement of Theorem 13). *An information-theoretic learner can (robustly) $\varepsilon$-learn the task of Construction 12 with parameter size $2\lambda$ and sample complexity $\Theta(\frac{\lambda}{\varepsilon})$. Moreover, an efficient learner can (robustly) $\varepsilon$-learn this task with parameter size $\alpha + \beta$ and sample complexity $\Theta(\frac{\alpha+\beta}{\varepsilon})$.*

Since the learning task of Theorem 13 has a finite hypothesis class, its learnability follows from the classical result of learning finite classes [Shalev-Shwartz and Ben-David, 2014]. Moreover, this can be done efficiently as this is a linear task. When it comes to learning *robust* functions, one can also use the result of Bubeck et al. [2019] for robustly learning finite classes.[17] The learner of Bubeck et al. [2019] simply uses the empirical-risk minimization, however this is done with respect to the *robust* empirical risk. This learner is not always polynomial-time, even if the (regular) risk minimization can be done efficiently. However, we would like to find *robust* learners also *efficiently*. The formal proof follows.

*Proof of Theorem 13.* Consider the set of functions $f_{s,s'} : \mathcal{X}_\lambda \to \mathcal{Y}_\lambda$ for all $s, s' \in \{0,1\}^\lambda$ as follows.

1. On input $x = (\widetilde{[u_1]}, \widetilde{[u_2]}, \widetilde{m}, \widetilde{\mathsf{d}})$, it invokes the error-correcting decoding algorithm on $\widetilde{[u_1]}$ and $\widetilde{[u_2]}$ to find $u_1$ and $u_2$.

2. It uses $\widetilde{\mathsf{d}} + f_2(s')|_{\mathsf{samp}(u_2)}$ to get an encoding $\widetilde{\mathsf{c}}$ of $m$.

3. It invokes the error-correcting decoding on $\widetilde{\mathsf{c}}$ to get $m$.

4. It outputs $\langle m, f_1(s)|_{\mathsf{samp}(u_1)} \rangle$.

One of the function $f_{s,s'}$ will (perfectly) robustly fit the distribution since all the encodings $[u_1], [u_2], \mathsf{Enc}(m)$ tolerates $(1-R)n/2$ perturbation. Since, there are $2^{2\lambda}$ such functions, by Theorem 19, we conclude that an information-theoretic learner can $\varepsilon$-learn this task with $2\lambda$ parameters and sample complexity $\Theta(\lambda/\varepsilon)$.

Note that an efficient learner might not be able to find such a function $f_{s,s'}$ as it requires inverting a pseudorandom generator. However, an efficient learner can still learn using more samples and parameters as follows. Consider the set of functions $f_{P,Q} : \mathcal{X}_\lambda \to \mathcal{Y}_\lambda$ for all $P \in \{0,1\}^\alpha$ and $Q \in \{0,1\}^\beta$ as follows.

1. On input $x = (\widetilde{[u_1]}, \widetilde{[u_2]}, \widetilde{m}, \widetilde{\mathsf{d}})$, it invokes the error-correcting decoding algorithm on $\widetilde{[u_1]}$ and $\widetilde{[u_2]}$ to find $u_1$ and $u_2$.

2. It uses $\widetilde{\mathsf{d}} + Q|_{\mathsf{samp}(u_2)}$ to get an encoding $\widetilde{\mathsf{c}}$ of $m$.

3. It invokes the error-correcting decoding on $\widetilde{\mathsf{c}}$ to get $m$.

4. It outputs $\langle m, P|_{\mathsf{samp}(u_1)} \rangle$.

---

[17]Results for infinite classes could be found in subsequent works Montasser et al. [2019].

Similarly, one of the function $f_{P,Q}$ will (perfectly) robustly fit the distribution since all the encodings $[u_1], [u_2], \mathsf{Enc}(m)$ tolerates $(1-R)n/2$ perturbation. Finding $f_{P,Q}$ that fits all the samples only requires linear operations and, hence, is efficiently learnable. As there are $2^{\alpha+\beta}$ such functions, again by Theorem 19, we conclude that an efficient learner can $\varepsilon$-learn this task with $\alpha + \beta$ parameters and sample complexity $\Theta((\alpha+\beta)/\varepsilon)$. To apply Theorem 19, we simply pretend that the hypothesis set is the larger set of $2^{\alpha+\beta}$ such functions, in which case our learner finds one of these $2^{\alpha+\beta}$ functions that perfectly matches with the training set with zero *robust* empirical risk. $\qquad\square$

**Theorem** (Restatement of Theorem 14). *Any efficient learner that outputs models with $\leqslant \alpha/2$ parameters cannot $\varepsilon$-learn $F_\lambda$ of Construction 12 for $\varepsilon < 1/3$.*

*Proof.* We start by defining another learning problem $F'_\lambda$. This learning problem is identical to $F_\lambda$ for $\mathcal{X}_\lambda, \mathcal{Y}_\lambda$, and $\mathcal{D}_\lambda$. However, $\mathcal{H}'_\lambda$ consists of all functions $h_P$ for all $P \in \{0,1\}^\alpha$, such that

$$h_P(x) = \left\langle m, \left( P\big|_{\mathsf{samp}_1(u_1)} \right) \right\rangle.$$

On a high level, our proof consists of two claims.

**Claim 29.** *Fix any distribution $D_{s'} \in \mathcal{D}_\lambda$. We consider a random hypothesis function $h_s$ and $h_P$, where $s \leftarrow \{0,1\}^\lambda$ and $P \leftarrow \{0,1\}^\alpha$. It holds that*

$$\mathop{\mathbf{E}}_{\mathcal{S} \leftarrow D_\lambda^n; f \leftarrow L(\mathcal{S}^{h_s}, \lambda)}[\mathrm{Risk}(h_s, D_\lambda, f)] \quad \approx_{\mathsf{negl}(\lambda)} \quad \mathop{\mathbf{E}}_{\mathcal{S} \leftarrow D_\lambda^n; f \leftarrow L(\mathcal{S}^{h_P}, \lambda)}[\mathrm{Risk}(h_P, D_\lambda, f)].$$

**Claim 30.** *For any learner $L$ (with an arbitrary sample complexity) with $\leqslant \alpha/2$ parameters, we have*

$$\mathop{\mathbf{E}}_{\mathcal{S} \leftarrow D_\lambda^n; f \leftarrow L(\mathcal{S}^{h_P}, \lambda)}[\mathrm{Risk}(h_P, D_\lambda, f)] > 3/8.$$

Note that, if both claims are correct, the theorem statement is true.

We first show Claim 29. Observe that, given the string $P$, one can compute the function $h_P(x)$ efficiently. Now, given a string $P$, which is either a pseudorandom string (i.e., $P \leftarrow f_1(U_\lambda)$) or a truly random string (i.e., $P \leftarrow \{0,1\}^\alpha$). Consider the following distinguisher

$$\left\{ \begin{array}{c} \mathcal{S} \leftarrow D_{s'}^n, \ f \leftarrow L(\mathcal{S}^{h_P}, \lambda), \ x \leftarrow D_{s'} \\ \text{Output } \mathbb{I}(h_P(x) = f(x)) \end{array} \right\}.$$

If $P$ is pseudorandom, the probability that the distinguisher outputing 1 is

$$\mathop{\mathbf{E}}_{\mathcal{S} \leftarrow D_\lambda^n; f \leftarrow L(\mathcal{S}^{h_s}, \lambda)}[\mathrm{Risk}(h_s, D_\lambda, f)];$$

if $P$ is truly random, the probability that the distinguisher outputs 1 is

$$\mathop{\mathbf{E}}_{\mathcal{S} \leftarrow D_\lambda^n; f \leftarrow L(\mathcal{S}^{h_P}, \lambda)}[\mathrm{Risk}(h_P, D_\lambda, f)].$$

Therefore, if Claim 29 does not hold, we break the pseudorandom property of the PRG.

It remains to prove Claim 30.

Since $P$ is sampled uniformly at random, we have $H_\infty(P) = \alpha$. Let $Z$ denote the random variable $L(\cdot, \lambda)$, i.e., the model learned by the learner. Since the learner's output model employs $\leqslant \alpha/2$ parameters, we have $\mathrm{Supp}(Z) \leqslant 2^{\alpha/2}$. And by Lemma 24, we must have

$$\widetilde{H}_\infty(P|Z) \geqslant \alpha/2.$$

Now, let us define the set[18]

$$\mathsf{Good} = \{z \in \mathrm{Supp}(Z) \ : \ H_\infty(P|Z=z) \leqslant \alpha/4\}.$$

Lemma 24 implies that

$$\Pr[Z \in \mathsf{Good}] \leqslant 2^{-\alpha/4} = \mathsf{negl}(\lambda).$$

---

[18]This set is called "good" as it is good for the learner.

In the rest of the analysis, we conditioned on the event that $Z \notin \mathsf{Good}$, which means $H_\infty(P|Z = z) > \alpha/4$. Now, let $x = ([u_1], [u_2], m, \mathsf{Enc}(m) + (f_2(s')\big|_{\mathsf{samp}_2(u_2)}))$ be the test sample. Since $P$ has min-entropy rate $> 1/4$, the property of the sampler (Lemma 11) guarantees that there exists a distribution $D$ such that

$$H_\infty(D) \geqslant \left(\frac{1}{4} - \kappa_1\right) \cdot k\ell > \frac{1}{5} \cdot k\ell$$

and

$$\mathsf{SD}\left((u_1, D)\,,\, (u_1, P\big|_{\mathsf{samp}_1(u_1)})\right) \leqslant \exp(-\Theta(\alpha\kappa_1)) + \exp\left(-n^{\kappa_2}\right) = \mathsf{negl}(\lambda).$$

Recall that $x = ([u_1], [u_2], m, \mathsf{Enc}(m) + (f_2(s')\big|_{\mathsf{samp}_2(u_2)}))$ and $y = h_P(x) = \langle m, (P\big|_{\mathsf{samp}_1(u_1)})\rangle$. Consequently,

$$\mathsf{SD}\left(((x, z), y)\,,\, ((x, z), U_{\{0,1\}})\right)$$
$$= \mathsf{SD}\left(((u_1, m, z), y)\,,\, ((u_1, m, z), U_{\{0,1\}})\right) \qquad \text{(as } u_2 \text{ is independent of } y)$$
$$\leqslant \mathsf{SD}\left(\left((u_1, m, z), \langle m, D\rangle\right)\,,\, ((u_1, m, z), U_{\{0,1\}})\right) + \mathsf{negl}(\lambda)$$
$$\qquad\qquad (\text{as } \mathsf{SD}\left(P_{\mathsf{samp}_1(u_1)}|Z = z\,,\, D\right) \leqslant \mathsf{negl}(\lambda))$$
$$\leqslant 2^{-k\ell/10} + \mathsf{negl}(\lambda) = \mathsf{negl}(\lambda). \qquad (\text{Theorem 25 and } H_\infty(D) > \tfrac{1}{5} \cdot k\ell)$$

Therefore, in the learner's view $(x, z)$, $y$ is statistically $\mathsf{negl}(\lambda)$-close to uniform. Therefore,

$$\mathop{\mathbf{E}}_{\mathcal{S} \leftarrow D_\lambda^n; f \leftarrow L(\mathcal{S}^{h_P}, \lambda)}[\mathrm{Risk}(h_P, D_\lambda, f)]$$
$$\geqslant \mathop{\mathrm{Pr}}_{\mathcal{S} \leftarrow D_\lambda^n; f \leftarrow L(\mathcal{S}^{h_P}, \lambda)}[z \in \mathsf{Good}]$$
$$+ \mathop{\mathrm{Pr}}_{\mathcal{S} \leftarrow D_\lambda^n; f \leftarrow L(\mathcal{S}^{h_P}, \lambda)}[z \notin \mathsf{Good}] \cdot \mathop{\mathbf{E}}_{\mathcal{S} \leftarrow D_\lambda^n; f \leftarrow L(\mathcal{S}^{h_P}, \lambda)}[\mathrm{Risk}(h_P, D_\lambda, f)|z \notin \mathsf{Good}]$$
$$\geqslant \mathsf{negl}(\lambda) + (1 - \mathsf{negl}(\lambda)) \cdot \left(\frac{1}{2} - \mathsf{negl}(\lambda)\right) > 3/8.$$

This shows Claim 30 and completes the proof of the theorem. $\qquad\qquad\square$

**Theorem** (Restatement of Theorem 15). *There exists some constant $c$ such that the following holds. In the presence of an adversary that may perturb $(1 - R)n/2$ symbols, any efficient learner for the task of Construction 12 that outputs a model with $c \cdot \beta/\log\lambda$ parameters cannot $\varepsilon$-robustly learn $F_\lambda$ for $\varepsilon < 1/3$.*

*Proof.* The high-level structure of the proof is similar to the proof of Theorem 14. We consider a new learning problem $F_\lambda'$ that has the same $\mathcal{X}_\lambda$, $\mathcal{Y}_\lambda$, and $\mathcal{H}_\lambda$. However, $\mathcal{D}_\lambda$ consists of all distribution $D_Q$ for all $Q \in \{0, 1\}^\beta$, where the distribution $D_Q$ is

$$D_Q = \left([u_1],\ [u_2],\ m,\ \mathsf{Enc}(m) + \left(Q\big|_{\mathsf{samp}_2(u_2)}\right)\right).$$

The proof consists of two claims.

**Claim 31.** *Fix a hypothesis $h_{s'} \in \mathcal{H}_\lambda$. We consider a random distribution over $\mathcal{D}_\lambda$ and $\mathcal{D}_\lambda'$. That is, $D_s$ and $D_Q$ are sampled with $s \leftarrow \{0, 1\}^\lambda$ and $Q \leftarrow \{0, 1\}^\beta$. It holds that*

$$\mathop{\mathbf{E}}_{\mathcal{S} \leftarrow D_s^n; f \leftarrow L(\mathcal{S}^{h_{s'}}, \lambda)}[\mathrm{Risk}(h_{s'}, D_s, f)] \quad \approx_{\mathsf{negl}(\lambda)} \quad \mathop{\mathbf{E}}_{\mathcal{S} \leftarrow D_Q^n; f \leftarrow L(\mathcal{S}^{h_{s'}}, \lambda)}[\mathrm{Risk}(h_{s'}, D_Q, f)].$$

**Claim 32.** *For any learner $L$ (with an arbitrary sample complexity) with $\leqslant c \cdot \beta/\log\lambda$ parameters, it holds that*

$$\mathop{\mathbf{E}}_{\mathcal{S} \leftarrow D_Q^n; f \leftarrow L(\mathcal{S}^{h_{s'}}, \lambda)}[\mathrm{Risk}(h_{s'}, D_Q, f)] > 3/8.$$

Note that these two claims prove the theorem. To see Claim 31, observe that given a string $Q$, we can sample efficiently from $D_Q$. Analogous to the proof of Claim 29, if Claim 31 does not hold, we may break the pseudorandom property of the PRG using this (efficient) learner $L$.

It remains to prove Claim 32.

Let $Z$ denote the random variable $L(\cdot, \lambda)$, i.e., the parameter of the learner. Since $Q$ is uniformly random, we have

$$H_\infty(Q|Z) \geqslant (1 - c/\log \lambda)\beta.$$

Let us define the set

$$\mathsf{Good} = \{z \in \mathrm{Supp}(Z) \ : \ H_\infty(Q|Z = z) \leqslant (1 - 2c/\log \lambda)\beta\}.$$

Lemma 24 implies that

$$\Pr\left[Z \in \mathsf{Good}\right] \leqslant 2^{-c\beta/\log \lambda} = \mathsf{negl}(\lambda).$$

In the rest of the analysis, we conditioned on the event that $Z \notin \mathsf{Good}$, which means $H_\infty(Q|Z = z) > (1 - 2c/\log \lambda)\beta$.

Now, we consider the following adversary $A$ that perturbs $(1 - R)n/2$ symbols. Given a test instance $(x, y)$, where

$$x = \left([u_1], \ [u_2], \ m, \ \mathsf{Enc}(m) + \left(Q\big|_{\mathsf{samp}_2(u_2)}\right)\right),$$

the adversary will do the following.

- Replace $m$ with a uniformly random string. This costs a budget of $Rn$.

- Samples a random subset $\mathcal{T} \subseteq \{1, 2, \ldots, n\}$ of size $(1 - 3R)n/2$. It adds noises to $\mathsf{Enc}(x) + \left(Q\big|_{\mathsf{samp}_2(u_2)}\right)$ at precisely those indices from $S$. This costs a budget of $(1 - 3R)n/2$.

  For simplicity, let us denote the distribution of this noise by $\rho$. That is, $\rho$ is a distribution over $\mathbb{F}_{2^\ell}^n$ such that it is 0 everywhere except for a random subset $\mathcal{T}$ and for those $i \in \mathcal{T}$, $\rho_i$ is uniformly random.

We now argue that the perturbed instance is statistically $\mathsf{negl}(\lambda)$-close to the distribution

$$\left([u_1], \ [u_2], \ U_{k\cdot\ell}, \ U_{n\cdot\ell}\right).$$

It suffices to prove that $\mathsf{Enc}(m) + \left(Q\big|_{\mathsf{samp}_2(u_2)}\right) + \rho$ is close to the uniform distribution. By Theorem 28, $\mathsf{Enc}(m) + \rho$ is $(1 - R)^{\frac{(1-3R)n}{2}}$-small-biased.

Furthermore, since $Q$ has min-entropy rate $> (1 - 2c/\log \lambda)$, the property of the sampler (Lemma 11) guarantees that there exists a distribution $D$ such that

$$H_\infty(D) \geqslant (1 - 2c/\log \lambda - \kappa_1) \cdot n\ell > (1 - 3c/\log \lambda) \cdot n\ell.$$

and

$$\mathsf{SD}\left((u_2, D), \ (u_2, Q|_{\mathsf{samp}_2(u_2)})\right) \leqslant \exp(-\Theta(\beta\kappa_1)) + \exp\left(-n^{\kappa_2}\right) = \mathsf{negl}(\lambda). \tag{1}$$

Finally, by Theorem 27, we have $(\mathsf{Enc}(m) + \rho) + D$ is

$$2^{\frac{3cn\ell}{2\log \lambda} - 1} \cdot (1 - R)^{\frac{(1-3R)n}{2}}$$

close to the uniform distribution. Observe that as long as

$$c < \frac{\log \lambda}{3\ell} \cdot (1 - 3R)\log(1/(1 - R)) = \Theta(1),$$

the closeness is negligible in $\lambda$. Overall,

$$\begin{aligned}
&\mathsf{SD}\left(\mathsf{Enc}(m) + Q|_{\mathsf{samp}_2(u_2)} + \rho, \ U_n\right) \\
&\leqslant \mathsf{SD}\left(\mathsf{Enc}(m) + Q|_{\mathsf{samp}_2(u_2)} + \rho, \ \mathsf{Enc}(m) + D + \rho\right) + \mathsf{SD}\left(\mathsf{Enc}(m) + D + \rho, \ U_n\right) \\
&\hspace{9cm} \text{(Triangle inequality)} \\
&\leqslant \mathsf{negl}(\lambda) + \mathsf{negl}(\lambda) = \mathsf{negl}(\lambda). \hspace{3cm} \text{(Equation 1 and Theorem 28)}
\end{aligned}$$

Therefore, given a test instance $x$ and the perturbed input $x'$, we have

$$\mathsf{SD}\left(((x',z),y)\,,\,((x',z),U_{\{0,1\}})\right)$$
$$\leqslant \mathsf{SD}\left(((u_1,u_2,U_{k\ell},U_{n\ell}),y)\,,\,((u_1,u_2,U_{k\ell},U_{n\ell}),U_{\{0,1\}})\right) + \mathsf{negl}(\lambda)$$
$$= \mathsf{SD}\left((u_1,\langle m,f_1(s')|_{\mathsf{samp}_1(u_1)}\rangle)\,,\,(u_1,U_{\{0,1\}})\right) + \mathsf{negl}(\lambda)$$
$$= \mathsf{negl}(\lambda).$$

Hence, when $z \notin \mathsf{Good}$, given a perturbed test input $x'$, the correct label $y$ is information-theoretically unpredicatable from the learner with $\mathsf{negl}(\lambda)$ advantage.

Putting everything together, we have

$$\underset{\mathcal{S}\leftarrow D_Q^n; f\leftarrow L(\mathcal{S}^{h_{s'}},\lambda)}{\mathbf{E}}\left[\mathrm{Risk}(h_{s'},D_Q,f)\right]$$
$$\geqslant \underset{\mathcal{S}\leftarrow D_Q^n; f\leftarrow L(\mathcal{S}^{h_{s'}},\lambda)}{\Pr}\left[z\in\mathsf{Good}\right]$$
$$+\underset{\mathcal{S}\leftarrow D_Q^n; f\leftarrow L(\mathcal{S}^{h_{s'}},\lambda)}{\Pr}\left[z\notin\mathsf{Good}\right]\cdot \underset{\mathcal{S}\leftarrow D_Q^n; f\leftarrow L(\mathcal{S}^{h_{s'}},\lambda)}{\mathbf{E}}\left[\mathrm{Risk}(h_{s'},D_Q,f)|z\notin\mathsf{Good}\right]$$
$$\geqslant \mathsf{negl}(\lambda) + (1-\mathsf{negl}(\lambda))\cdot\left(\frac{1}{2}-\mathsf{negl}(\lambda)\right) > 3/8.$$

This completes the proof of the claim and the entire theorem. $\qquad\square$

## C Computationally robust learning could need fewer parameters

In this section, we formally prove Part 2 of Theorem 1. Our construction of the learning problem is formally presented in Construction 16.

We note that the instance size is (approximately) $\Theta(n\cdot\ell) = \Theta(n\cdot\log\lambda)$. We shall prove two properties of this construction. In Theorem 17, we establish the *upper bound* of learnability with few parameters under efficient (polynomial-time) attacks. Later, in Theorem 18, we establish the lower bound of the number of parameters when the attacker is unbounded.

In the rest of this section, we formally prove these theorems.

**Theorem** (Restatement of Theorem 17). *For the learning task of Construction 16, there is an efficient learner (with 0 sample complexity) that outputs a model with no parameter and $\mathsf{negl}(\lambda)$-robustly learns $F_\lambda$ against efficient adversaries of budget $(1-\sqrt{R})n$.*

*Proof.* The learner is defined as follows. On input a perturbed instance $x' = (\widetilde{[u]},\widetilde{[v]},\widetilde{[\mathsf{vk}]},\widetilde{\mathsf{c}},\widetilde{\mathsf{d}})$, it does the following:

1. Invoke the error-correction algorithm to recover $\mathsf{vk}$.

2. Invoke the list-decoding algorithm on $\widetilde{\mathsf{c}}$ to find a list of message/signature $(b_i,\sigma_i)$ pairs.

3. Run the verifier to find any valid message/signature pair $(b^*,\sigma^*)$ and output $b^*$. If no such pair exists, output a random bit, and if there are more than one such pair, pick one arbitrarily.

Observe that the learner can always recover the correct $\mathsf{vk}$ since the encoding $[\mathsf{vk}]$ tolerates $(1-\sqrt{R})n$ errors.

Next, suppose the original instance is $([u],[v],[\mathsf{vk}],\mathsf{LEnc}(b,\mathsf{Sign}(\mathsf{sk},b)))$. Then, $(b,\mathsf{Sign}(\mathsf{sk},b))$ is always in the list of message/signature pairs output by the list-decoding algorithm. This is due to that $\mathsf{LEnc}(b,\mathsf{Sign}(\mathsf{sk},b))$ is $(1-\sqrt{R})n$-close to the perturbed encoding $\widetilde{\mathsf{c}}$ and the list decoding algorithm outputs all such messages whose encoding is $(1-\sqrt{R})n$-close to the perturbed one.

Finally, fix any distribution $D_s$. It must hold that, with $1-\mathsf{negl}(\lambda)$ probability, there does not exist a valid message/signature pair where the message is $1-b$. If this does not hold, one may utilize this learning adversary $A$ to break the unforgeability of the signature scheme as follows: on

input the verification key $\mathsf{vk}$ and a valid message/signature $(b, \mathsf{Sign}(\mathsf{sk}, b))$, the signature adversary samples the test instance and feed it to the adversary $A$, obtaining a perturbed instance $x' = (\widetilde{[u]}, \widetilde{[v]}, \widetilde{[\mathsf{vk}]}, \widetilde{c}, \widetilde{d})$.[19] The signature adversary uses the same procedure as the efficient learner to recover a list of message/signature pairs. If there is a valid message/signature pair with message $1 - b$, clearly, the signature adversary breaks the unforgeability of the signature scheme. Since the signature scheme is $\mathsf{negl}(\lambda)$-secure, it must hold that, with $1 - \mathsf{negl}(\lambda)$ probability, there does not exist a valid message/signature pair where the message is $1 - b$.

Consequently, this efficient learner outputs the correct label $b$ with $1 - \mathsf{negl}(\lambda)$ probability. Thus, for all efficient adversary $A$,

$$\mathop{\mathbf{E}}_{f \leftarrow L(\emptyset, \lambda)}[\mathrm{Risk}_{d,r}(h, D_s, f)] = \mathsf{negl}(\lambda),$$

and this finishes the proof. $\qquad\square$

**Theorem** (Restatement of Theorem 18). *Consider the learning task of Construction 16. Then, for computationally unbounded adversaries, any information-theoretic learner with $\leqslant \alpha/2$ parameters cannot $\varepsilon$-robustly learn $F_\lambda$ for $\varepsilon < 1/3$.*

*Proof.* We sample $s$ uniformly at random from $\{0,1\}^\alpha$ and prove that

$$\mathop{\mathbf{E}}_{\mathcal{S} \leftarrow D_s^n; f \leftarrow L(\mathcal{S}^h, \lambda)}[\mathrm{Risk}_{d,r}(h, D_s, f)] > 1/3.$$

The proof is similar to the proof of Theorem 14.

Let $Z$ denote $L(\cdot, \lambda)$, i.e., the parameters of the model output by the learner. Given a test instance $x = ([u], [v], [\mathsf{vk}], \mathsf{LEnc}(b, \mathsf{Sign}(\mathsf{sk}, b)), [b + \langle v, s|_{\mathsf{samp}(u)} \rangle])$, we first prove the following claim.

**Claim 33.** *With overwhelming probability over $Z$,*

$$\left( Z, ([u], [v], [\mathsf{vk}], \mathsf{LEnc}(b, \mathsf{Sign}(\mathsf{sk}, b)), [b + \langle v, s|_{\mathsf{samp}(u)} \rangle]) \right)$$

$$\approx_{\mathsf{negl}(\lambda)} \left( Z, ([u], [v], [\mathsf{vk}], \mathsf{LEnc}(b, \mathsf{Sign}(\mathsf{sk}, b)), [U_{\{0,1\}}]) \right).$$

That is, the learner cannot distinguish the two distributions given $Z$.

First, we have $\widetilde{H}_\infty(s|Z) \geqslant \alpha/2$. Define the set

$$\mathsf{Good} = \{z \ : \ H_\infty(s|Z = z) \leqslant \alpha/4\}.$$

By Lemma 24, $\Pr[Z \in \mathsf{Good}] \leqslant 2^{-\alpha/4} = \mathsf{negl}(\lambda)$. For the rest of the analysis, we conditioned on $Z \notin \mathsf{Good}$. Since $H_\infty(s|Z = z) > \alpha/4$, by the property of the sampler (Lemma 11), there exists a distribution $D$ such that

$$H_\infty(D) \geqslant \left( \frac{1}{4} - \kappa_1 \right) \alpha > \frac{1}{5}\alpha$$

and

$$\mathsf{SD}\left( (u, s|_{\mathsf{samp}(u)}) , (u, D) \right) \leqslant \exp(-\Theta(\beta \kappa_1)) + \exp(-n^{\kappa_2}) = \mathsf{negl}(\lambda). \qquad (2)$$

Finally, by Theorem 25, we have

$$\mathsf{SD}\left( (v, \langle v, s|_{\mathsf{samp}(u)} \rangle) , (v, U_{\{0,1\}}) \right)$$
$$\leqslant \mathsf{SD}\left( (v, \langle v, s|_{\mathsf{samp}(u)} \rangle) , (v, \langle v, D \rangle) \right) + \mathsf{SD}\left( (v, \langle v, D \rangle) , (v, U_{\{0,1\}}) \right) \quad \text{(Triangle inequality)}$$
$$\leqslant \mathsf{negl}(\lambda) + \mathsf{negl}(\lambda). \qquad\qquad \text{(Equation 2 and Theorem 25)}$$

This completes the proof of Claim 33.

Now, consider the following adversary $A$ that perturbs $n/2$ symbols and does the following. (Observe that $n/2 < (1 - \sqrt{R})n$ for $R < 1/4$ and, hence, the adversary is within budget.)

---

[19]Note that the adversary can efficiently sample from $D_s$ as the $(\mathsf{vk}, \mathsf{sk})$ pairs for every instance are independent.

1. $A$ decodes $\mathsf{LEnc}(b, \mathsf{Sign}(\mathsf{sk}, b))$ to find $b$. Let $\bar{b} = 1 - b$. It forges a valid signature $\sigma = \mathsf{Sign}(\mathsf{sk}, \bar{b})$ and encode it $\mathsf{LEnc}(\bar{b}, \sigma)$.

2. Now, for a random subset $\mathcal{T} \subseteq \{1, 2, \ldots, n\}$ of size $|\mathcal{T}| = n/2$, $A$ replaces $\mathsf{LEnc}(b, \mathsf{Sign}(\mathsf{sk}, b))$ with $\mathsf{LEnc}(\bar{b}, \sigma)$ on those $i \in \mathcal{T}$. Then, after perturbation, the string $\mathsf{LEnc}(b, \mathsf{Sign}(\mathsf{sk}, b))$ becomes a *random string* that has Hamming distance exactly $n/2$ from both $(0, \mathsf{Sign}(\mathsf{sk}, 0))$ and $(1, \mathsf{Sign}(\mathsf{sk}, 1))$. Let us call this distribution $X$. Note that $X$ is independent of $b$.

Therefore, after the perturbation, the perturbed instance is statistically close to
$$([u], [v], [\mathsf{vk}], X, [U_{\{0,1\}}]),$$
which is independent of $b$. Hence, the learner's output will not agree with $b$ with probability $\geqslant 1/2 - \mathsf{negl}(\lambda)$. Putting everything together, we have
$$\mathop{\mathbf{E}}_{\mathcal{S} \leftarrow D_s^n; f \leftarrow L(\mathcal{S}^h, \lambda)} [\mathrm{Risk}_{d,r}(h, D_s, f)]$$
$$\geqslant \mathop{\Pr}_{\mathcal{S} \leftarrow D_s^n; f \leftarrow L(\mathcal{S}^h, \lambda)} [z \in \mathsf{Good}]$$
$$+ \mathop{\Pr}_{\mathcal{S} \leftarrow D_s^n; f \leftarrow L(\mathcal{S}^h, \lambda)} [z \notin \mathsf{Good}] \cdot \mathop{\mathbf{E}}_{\mathcal{S} \leftarrow D_s^n; f \leftarrow L(\mathcal{S}^h, \lambda)} [\mathrm{Risk}_{d,r}(h, D_s, f) | z \notin \mathsf{Good}]$$
$$\geqslant \mathsf{negl}(\lambda) + (1 - \mathsf{negl}(\lambda)) \cdot \left( \frac{1}{2} - \mathsf{negl}(\lambda) \right) > 1/3. \qquad \square$$

## D  Missing Proofs

### D.1  Proof of Theorem 25

The theorem follows from the following derivation.
$$\mathsf{SD}\left( \left( Y, \langle X, Y \rangle \right), \left( Y, U_{\{0,1\}} \right) \right)$$
$$= \mathop{\mathrm{E}}_{y \leftarrow Y} \left[ \mathsf{SD}\left( \langle X, y \rangle, U_{\{0,1\}} \right) \right]$$
$$= \frac{1}{2} \cdot \mathop{\mathrm{E}}_{y \leftarrow Y} \left[ |\Pr[\langle X, y \rangle = 0] - \Pr[\langle X, y \rangle = 1]| \right]$$
$$\leqslant \frac{1}{2} \cdot \sqrt{ \mathop{\mathrm{E}}_{y \leftarrow Y} \left[ (\Pr[\langle X, y \rangle = 0] - \Pr[\langle X, y \rangle = 1])^2 \right] } \qquad \text{(Jensen's inequality)}$$
$$= \frac{1}{2} \cdot \sqrt{ \mathop{\mathrm{E}}_{y \leftarrow Y} \left[ \mathop{\Pr}_{x, x' \leftarrow X} [\langle x, y \rangle = \langle x', y \rangle] - \mathop{\Pr}_{x, x' \leftarrow X} [\langle x, y \rangle \neq \langle x', y \rangle] \right] }$$
$$= \frac{1}{2} \cdot \sqrt{ \mathop{\Pr}_{x, x' \leftarrow X} \left[ \mathop{\Pr}_{y \leftarrow Y} [\langle x - x', y \rangle = 0] - \mathop{\Pr}_{y \leftarrow Y} [\langle x - x', y \rangle = 1] \right] }$$
$$= \frac{1}{2} \cdot \sqrt{ \mathop{\Pr}_{x, x' \leftarrow X} [x = x'] \cdot \frac{1}{2} } \qquad \text{(when } x \neq x', \text{ the inner term is always 0)}$$
$$= \frac{1}{2} \cdot \sqrt{ \frac{1}{2} \cdot \sum_\omega (\Pr[X = \omega])^2 }$$
$$\leqslant \frac{1}{2} \cdot \sqrt{ \frac{1}{2} \cdot \sum_\omega \Pr[X = \omega] \cdot 2^{-H_\infty(X)} } \qquad \text{(By definition of min-entropy)}$$
$$= \frac{1}{2} \cdot \sqrt{ 2^{-H_\infty(X) - 1} } \leqslant \varepsilon$$

### D.2  Proof of Theorem 27

Dodis and Smith Dodis and Smith [2005] proved this theorem for $\mathbb{F}_2$. We are simply revising their proof for the field $\mathbb{F}_{2^\ell}$. Within this proof, we shall use $\mathbb{F}$ for $\mathbb{F}_{2^\ell}$. We need the following claims.

**Claim 34** (Parseval's identity). $\sum_\alpha \text{bias}(X, \alpha)^2 = |\mathbb{F}|^n \cdot \sum_\omega (\Pr[X = \omega])^2$.

*Proof.* Observe that

$$\sum_\alpha \text{bias}(X, \alpha)^2$$

$$= \sum_\alpha \left( \operatorname*{E}_{x \leftarrow X} \left[ (-1)^{\text{Tr}(\langle x, \alpha \rangle)} \right] \right)^2$$

$$= \sum_\alpha \operatorname*{E}_{x, x' \leftarrow X} \left[ (-1)^{\text{Tr}(\langle x, \alpha \rangle)} \cdot (-1)^{\text{Tr}(\langle x', \alpha \rangle)} \right]$$

$$= \sum_\alpha \operatorname*{E}_{x, x' \leftarrow X} \left[ (-1)^{\text{Tr}(\langle x + x', \alpha \rangle)} \right] \qquad \text{(Since the trace map is additive)}$$

$$= \operatorname*{E}_{x, x' \leftarrow X} \left[ \sum_\alpha (-1)^{\text{Tr}(\langle x + x', \alpha \rangle)} \right]$$

$$= |\mathbb{F}|^n \operatorname*{Pr}_{x, x' \leftarrow X} [x = x'].$$

Here, we use the fact that, when $x \neq x'$, the inner term is 0 as the trace map maps half of the field to 0 and the other half to 1.

Note that the last line is exactly equal to

$$|\mathbb{F}|^n \cdot \sum_\omega (\Pr[X = \omega])^2. \qquad \square$$

**Claim 35** (Bias of Convolusion is product of bias). $\text{bias}(X + Y, \alpha) = \text{bias}(X, \alpha) \cdot \text{bias}(Y, \alpha)$.

*Proof.* Observe that

$$\text{bias}(X + Y, \alpha)$$

$$= \sum_\omega \Pr[X + Y = \omega] \cdot (-1)^{\text{Tr}(\langle \omega, \alpha \rangle)}$$

$$= \sum_\omega \sum_{\omega'} \Pr[X = \omega'] \Pr[Y = \omega - \omega'] \cdot (-1)^{\text{Tr}(\langle \omega, \alpha \rangle)}$$

$$= \sum_{\omega''} \sum_{\omega'} \Pr[X = \omega'] \Pr[Y = \omega''] \cdot (-1)^{\text{Tr}(\langle \omega' + \omega'', \alpha \rangle)}$$

$$= \left( \sum_{\omega'} \Pr[X = \omega'] \cdot (-1)^{\text{Tr}(\langle \omega', \alpha \rangle)} \right) \cdot \left( \sum_{\omega''} \Pr[Y = \omega''] \cdot (-1)^{\text{Tr}(\langle \omega'', \alpha \rangle)} \right)$$

$$= \text{bias}(X, \alpha) \cdot \text{bias}(Y, \alpha). \qquad \square$$

Given these two claims, we prove the theorem as follows.

$$\text{SD}(X + Y, U_{\mathbb{F}^n})$$

$$= \frac{1}{2} \cdot \sum_\omega |\Pr[X + Y = \omega] - \Pr[U_{\mathbb{F}^n} = \omega]|$$

$$\leqslant \frac{1}{2} \cdot \sqrt{|\mathbb{F}|^n \cdot \sum_\omega (\Pr[X + Y = \omega] - \Pr[U_{\mathbb{F}^n} = \omega])^2} \qquad \text{(Cauchy-Schwartz)}$$

$$= \frac{1}{2} \cdot \sqrt{\sum_\alpha (\text{bias}(X + Y, \alpha) - \text{bias}(U_{\mathbb{F}^n}, \alpha))^2} \qquad \text{(Parseval)}$$

$$= \frac{1}{2} \cdot \sqrt{\sum_{\alpha \neq 0^n} \text{bias}(X + Y, \alpha)^2} \qquad \text{(Since bias}(U_{\mathbb{F}^n}, \alpha) = 0 \text{ for all } \alpha \neq 0^n.)$$

$$= \frac{1}{2} \cdot \sqrt{\sum_{\alpha \neq 0^n} \mathsf{bias}(X, \alpha)^2 \cdot \mathsf{bias}(Y, \alpha)^2} \qquad \text{(By Claim 35)}$$

$$\leqslant \frac{\varepsilon}{2} \cdot \sqrt{\sum_{\alpha \neq 0^n} \mathsf{bias}(X, \alpha)^2} \qquad \text{(Since } Y \text{ is small-biased)}$$

$$\leqslant \frac{\varepsilon}{2} \cdot \sqrt{|\mathbb{F}|^n \sum_{\omega} (\Pr[X = \omega])^2} \qquad \text{(Parseval)}$$

$$= \frac{\varepsilon}{2} \cdot \sqrt{|\mathbb{F}|^n \sum_{\omega} \Pr[X = \omega] \cdot 2^{-H_\infty(X)}} \qquad \text{(Definition of min-entropy)}$$

$$= \frac{\varepsilon}{2} \cdot \sqrt{|\mathbb{F}|^n \cdot 2^{-H_\infty(X)}}$$

$$= 2^{\frac{n\ell - k}{2} - 1} \cdot \varepsilon$$

### D.3   Proof of Theorem 28

We divide all possible linear tests $\alpha$ into two cases.

- **Small linear tests are fooled by RS code.** We say that $\alpha$ is a small linear test if $|\{i \colon \alpha_i \neq 0\}| \leqslant Rn$. By Fact 22, a random codeword projects onto any $\leqslant Rn$ coordinates is always a uniform distribution. Hence, $\langle D, \alpha \rangle$ is always uniform. Consequently, $\mathsf{bias}(D, \alpha) = 0$ for all small linear test.

- **Large linear tests are fooled by the noise.** Suppose $\alpha$ is such that $|\mathcal{T}| > Rn$, where $\mathcal{T} = \{i \colon \alpha_i \neq 0\}$. Observe that $\mathcal{S}$ is a random subset of size $s$ and $\mathcal{T}$ is a fixed set of size $> Rn$. Clearly, $\mathcal{S} \cap \mathcal{T} = \emptyset$ happens with probability $\leqslant (1 - R)^s$. Now, conditioned on the event that $\mathcal{S} \cap \mathcal{T} \neq \emptyset$, we again have $\langle D, \alpha \rangle$ is a uniform distribution (because of the random noise). Consequently, for large $\alpha$, we have $\mathsf{bias}(D, \alpha) \leqslant (1 - R)^s$.

Therefore, for all possible $\alpha$, $\mathsf{bias}(D, \alpha)$ is small. Hence, the theorem follows.