# OpenReview forum: "Overparameterization from Computational Constraints"
_NeurIPS.cc/2022/Conference — NeurIPS 2022 Accept_

### Official Review · Reviewer_Sv39 · 2022-07-10

**Rating:** 5
**Confidence:** 1
**Soundness:** 3 good
**Presentation:** 2 fair
**Contribution:** 3 good

**Summary:**

This work studies whether there exist any learning tasks that are computationally bounded robust learning need to have more model parameters than non-robust learning; and whether such tasks can be done with fewer model parameters for polynomial-time adversaries compared with information-theoretic adversaries. The author gives positive answers to both of the questions by leveraging cryptography and coding theory.

**Questions:**

1. Missing formal definition of $L(\cdot,\lambda)$.
2. In the main paper top of page 6 you discuss theorem 32, which is a restatement of theorem 13, and would be good to state theorem 13 before discussing it. In the statement of construction 10, you mention theorem 20 which also does not appear in the main paper.
3. Personally I did not find it’s helpful to give construction 10 and 14 in detail in the main paper.  It’d be helpful to combine theorem 11, 12, 13, 15, 16 with their discussion or comments instead of isolating them separately.
4. It seems to me that the argument of computational perspective on page 5 line 154-164 has a similar idea as the robustness on page 6 line 169-177, and I did not find any arguments regarding why $\alpha<\beta$, but this should be an important criterion in theorem 1. Can you claim what would happen if I replace $\beta$ with $\alpha$ in argument line 169-177?


**Limitations:**

Yes

**Strengths And Weaknesses:**

The questions asked by the work are interesting and novel to the best of my knowledge. Such studies would be helpful for the robust adversarial learning community.
The writing needs to have more structure improvement. However, since I’m not familiar with cryptography and coding theory, I’m not able to judge whether the technical idea is novel or sound.

---

> ### Author Response · Authors · 2022-08-01
> **Response to Reviewer Sv39**
>
> Thanks for pointing out the issues regarding missing definitions and the ordering of theorems. We shall be happy to fix them and make appropriate modifications to ensure that the theorems we cite appear in the main paper.
>
> Our rationale for including the formal constructions and theorem statement in the main body is to give a formal treatment of the intuitive claims we make during the high-level overview.
>
> The techniques we use in non-robust learning (pg 5, 154-164) and robust learning (pg 6, 169-177) are indeed similar. Note that we have the freedom to pick any alpha and beta (as an arbitrary polynomial in lambda) we want. Our construction chooses to pick beta >> alpha. This construction demonstrates our point that there is a learning problem such that a robust efficient learner needs more parameters (i.e., alpha + beta) than a non-robust efficient learner (i.e., alpha). One could also imagine the case where beta < alpha. However, it would give a learning problem where the efficient robust learner and efficient non-robust learner require a similar number of parameters, which is not very interesting to us.

---

### Official Review · Reviewer_WAyy · 2022-07-12

**Rating:** 7
**Confidence:** 3
**Soundness:** 4 excellent
**Presentation:** 3 good
**Contribution:** 3 good

**Summary:**

This paper studies the computational benefits of overparametrization in standard and robust learning. The paper suggests that there are learning tasks where overparametrization is needed from a computationally efficient, standard and robust, learning.

More specifically, as I understand it,  the paper shows in Theorem 1 (Part 1) that there are learning tasks where “efficient” standard learning and “efficient” robust learning require polynomial (in the input/instance size) overparameterization.

In Theorem 1 Part 2, the paper shows that there are robust learning tasks such that there are efficient underparameterized learners against computationally bounded adversaries, but overparamertization is needed when the adversary is computationally unbounded (even for information-theoretically efficient robust learners).

The result indicates that there is a statistical-computational tradeoff between efficient robust learning and inefficient robust learning.


**Questions:**

In Theorem 1 Part 1, by checking Theorem 11, it seems that the sample complexity is also polynomially larger. I wonder if it is possible to obtain a result where the efficient learner only needs overparamterization but no increase in sample complexity compared to the information-theoretic learner.


**Limitations:**

 It might be good to reflect on the results of the paper, and discuss their practical implications, and any future directions that might be good to address from this work.


**Strengths And Weaknesses:**

I find the main result of the paper interesting. The gaps established in this work are polynomial (as opposed to exponential), but I think that’s still interesting from a practical perspective. The paper is rigorous and formal in its treatment of the definition of “efficient” learning, which is nice. The paper is clear and well-written.

---

> ### Author Response · Authors · 2022-08-01
> **Response to Reviewer WAyy**
>
> For theorem 11, the reviewer is correct that the efficient learner does need (multiplicatively) polynomially more samples than an information-theoretic learner. The construction we have in this paper does not give such an example where the efficient learner only requires more parameters but the same amount of samples. We believe this is an excellent question and exciting future work.

---

> > ### Comment · Reviewer_WAyy · 2022-08-08
> > **Thank you.**
> >
> > Thank you for addressing my question. I strongly suggest that the authors attempt to discuss practical implications in future revisions of their work.

---

> > > ### Author Response · Authors · 2022-08-09
> > > **new section added**
> > >
> > > Dear reviewer WAyy,
> > >
> > > Thank you for your suggestion. We added a new discussion in appendix E reflecting on implications and limitations of our work. We included this in appendix due to space limitations. We will include this in the main body of our paper in the potential camera-ready version, which allows for an additional page.

---

### Official Review · Reviewer_7oiG · 2022-07-17

**Rating:** 6
**Confidence:** 1
**Soundness:** 2 fair
**Presentation:** 2 fair
**Contribution:** 2 fair

**Summary:**

The paper considers the tasks of adversarial robustness and explores the difference between omniscient adversary, and the computationally bounded adversary. It constructs several learning tasks demonstrating the difference  between informatic theoretic and efficient learning.

**Questions:**

Do the results work also for infinite (and in particular, uncountable) domains? Such as learning something from real-numbered observations? Specially the number of parameters needs to be reconsidered here due to space filling curves.

I'm sorry for such an uninformative review; it did not match my knowledge and I don't have the time to understand the relevant context.

**Limitations:**

-

**Strengths And Weaknesses:**

I did not find anything wrong with the paper, but I haven't understood its arguments either due to my limited knowledge of this area.

---

> ### Author Response · Authors · 2022-08-01
> **Response to Reviewer 7oiG**
>
> In this work, we consider the simplest setting of binary output and construct such a learning problem where there is an arbitrary polynomial gap in the number of parameters that an efficient learner and an information-theoretic learner require. One can extend it to more sophisticated settings of any finite output range, particularly real numbers with bounded precision. However, if one considers real numbers with infinite precision, it is unclear what it means for an algorithm to be computationally efficient (as it needs to take an infinitely long input). Hence, we do not consider it in this work.
>
> We are not sure if we completely understand what the reviewer means by “ the number of parameters needs to be reconsidered here due to space filling curves.” We are happy to clarify more if the reviewer could elaborate or has more questions.

---

> > ### Comment · Reviewer_7oiG · 2022-08-03
> > **Respone**
> >
> > Thank you for your response.
> >
> > I did not realize that the efficient learning of real parameters is "undefined". It sort of answers the question; the point was that 1 real parameter is indistinguishable from any countable set of real parameters (bijection between $\mathbb{R}$ nd $\mathbb{R}^\omega$).

---

> > > ### Author Response · Authors · 2022-08-09
> > > **Thank you.**
> > >
> > > Dear reviewer 7oiG,
> > >
> > > Thank you for raising this interesting point. We added a discussion on this in appendix E. There, we mention that our work does not consider real valued parameters with infinite precision because computationally bounded machines are not able to store or read them. We will move this discussion to the main body in the camera-ready version.

---

### Author Response · Authors · 2022-08-05
**To all reviewers**

We appreciated the thoughtful review, and are available to answer any questions.

---

### Meta-Review · Area_Chair_Nq4A · 2022-08-23

**Recommendation:** Accept
**Confidence:** Less certain

**Metareview:**

This paper studies the computational benefits of overparametrization in standard and robust learning. The paper suggests that there are learning tasks where overparametrization is needed from a computationally efficient, standard and robust, learning.
In Theorem 1 Part 2, the paper shows that there are robust learning tasks such that there are efficient underparameterized learners against computationally bounded adversaries, but overparamertization is needed when the adversary is computationally unbounded (even for information-theoretically efficient robust learners).
The result indicates that there is a statistical-computational tradeoff between efficient robust learning and inefficient robust learning.

The referees are leaning towards acceptance and that is also my recommendation,

**Award:**

No

---

### Decision · Program_Chairs · 2022-09-14

Accept